# On the Universality of Rotation Equivariant Point Cloud Networks

**Nadav Dym**
Duke University
nadavdym@gmail.com

**Haggai Maron**
NVIDIA Research
hmaron@nvidia.com

## Abstract

Learning functions on point clouds has applications in many fields, including computer vision, computer graphics, physics, and chemistry. Recently, there has been a growing interest in neural architectures that are invariant or equivariant to all three shape-preserving transformations of point clouds: translation, rotation, and permutation. In this paper, we present a first study of the approximation power of these architectures. We first derive two sufficient conditions for an equivariant architecture to have the universal approximation property, based on a novel characterization of the space of equivariant polynomials. We then use these conditions to show that two recently suggested models (Thomas et al., 2018; Fuchs et al., 2020) are universal, and for devising two other novel universal architectures.

## 1 Introduction

Designing neural networks that respect data symmetry is a powerful approach for obtaining efficient deep models. Prominent examples being convolutional networks which respect the translational invariance of images, graph neural networks which respect the permutation invariance of graphs (Gilmer et al., 2017; Maron et al., 2019b), networks such as (Zaheer et al., 2017; Qi et al., 2017a) which respect the permutation invariance of sets, and networks which respect 3D rotational symmetries (Cohen et al., 2018; Weiler et al., 2018; Esteves et al., 2018; Worrall & Brostow, 2018; Kondor et al., 2018a).

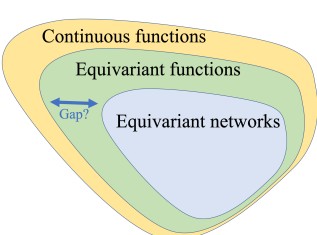

While the expressive power of equivariant models is reduced by design to include only equivariant functions, a desirable property of equivariant networks is universality: the ability to approximate any continuous equivariant function. This is not always the case: while convolutional networks and networks for sets are universal (Yarotsky, 2018; Segol & Lipman, 2019), popular graph neural networks are not (Xu et al., 2019; Morris et al., 2018).

In this paper, we consider the universality of networks that respect the symmetries of 3D point clouds: translations, rotations, and permutations. Designing such networks is a popular paradigm in recent years (Thomas et al., 2018; Fuchs et al., 2020; Poulenard et al., 2019; Zhao et al., 2019). While there have been many works on the universality of permutation invariant networks (Zaheer et al., 2017; Maron et al., 2019c; Keriven & Peyré, 2019), and a recent work discussing the universality of rotation equivariant networks (Bogatskiy et al., 2020), this is a first paper which discusses the universality of networks which combine rotations, permutations and translations.

We start the paper with a general, architecture-agnostic, discussion, and derive two sufficient conditions for universality. These conditions are a result of a novel characterization of equivariant polynomials for the symmetry group of interest. We use these conditions in order to prove universality of the prominent *Tensor Field Networks* (TFN) architecture (Thomas et al., 2018; Fuchs et al., 2020). The following is a weakened and simplified statement of Theorem 2 stated later on in the paper:

**Theorem** (Simplification of Theorem 2). *Any continuous equivariant function on point clouds can be approximated uniformly on compact sets by a composition of TFN layers.*

We use our general discussion to prove the universality of two additional equivariant models: the first is a simple modification of the TFN architecture which allows for universality using only low dimensional filters. The second is a minimal architecture which is based on tensor product representations, rather than the more commonly used irreducible representations of $SO(3)$. We discuss the advantages and disadvantages of both approaches.

To summarize, the contributions of this paper are: (1) A general approach for proving the universality of rotation equivariant models for point clouds; (2) A proof that two recent equivariant models (Thomas et al., 2018; Fuchs et al., 2020) are universal; (3) Two additional simple and novel universal architectures.

## 2 PREVIOUS WORK

**Deep learning on point clouds.** (Qi et al., 2017a; Zaheer et al., 2017) were the first to apply neural networks directly to the raw point cloud data, by using pointwise functions and pooling operations. Many subsequent works used local neighborhood information (Qi et al., 2017b; Wang et al., 2019b; Atzmon et al., 2018). We refer the reader to a recent survey for more details (Guo et al., 2020). In contrast with the aforementioned works which focused solely on permutation invariance, more related to this paper are works that additionally incorporated invariance to rigid motions. (Thomas et al., 2018) proposed Tensor Field Networks (TFN) and showed their efficacy on physics and chemistry tasks.(Kondor et al., 2018b) also suggested an equivariant model for continuous rotations. (Li et al., 2019) suggested models that are equivariant to discrete subgroups of $SO(3)$. (Poulenard et al., 2019) suggested an invariant model based on spherical harmonics. (Fuchs et al., 2020) followed TFN and added an attention mechanism. Recently, (Zhao et al., 2019) proposed a quaternion equivariant point capsule network that also achieves rotation and translation invariance.

**Universal approximation for invariant networks.** Understanding the approximation power of invariant models is a popular research goal. Most of the current results assume that the symmetry group is a permutation group. (Zaheer et al., 2017; Qi et al., 2017a; Segol & Lipman, 2019; Maron et al., 2020; Serviansky et al., 2020) proved universality for several $S_n$-invariant and equivariant models. (Maron et al., 2019b;a; Keriven & Peyré, 2019; Maehara & NT, 2019) studied the approximation power of high-order graph neural networks. (Maron et al., 2019c; Ravanbakhsh, 2020) targeted universality of networks that use high-order representations for permutation groups(Yarotsky, 2018) provided several theoretical constructions of universal equivariant neural network models based on polynomial invariants, including an $SE(2)$ equivariant model. In a recent work (Bogatskiy et al., 2020) presented a universal approximation theorem for networks that are equivariant to several Lie groups including $SO(3)$. The main difference from our paper is that we prove a universality theorem for a more complex group that besides rotations also includes translations and permutations.

## 3 A FRAMEWORK FOR PROVING UNIVERSALITY

In this section, we describe a framework for proving the universality of equivariant networks. We begin with some mathematical preliminaries:

### 3.1 MATHEMATICAL SETUP

An action of a group $G$ on a real vector space $W$ is a collection of maps $\rho(g) : W \to W$ defined for any $g \in G$, such that $\rho(g_1) \circ \rho(g_2) = \rho(g_1 g_2)$ for all $g_1, g_2 \in G$, and the identity element of $G$ is mapped to the identity mapping on $W$. We say $\rho$ is a *representation* of $G$ if $\rho(g)$ is a linear map for every $g \in G$. As is customary, when it does not cause confusion we often say that $W$ itself is a representation of $G$ .

In this paper, we are interested in functions on point clouds. Point clouds are sets of vectors in $\mathbb{R}^3$ arranged as matrices:

$$X = (x_1, \ldots, x_n) \in \mathbb{R}^{3 \times n}.$$

Many machine learning tasks on point clouds, such as classification, aim to learn a function which is invariant to rigid motions and relabeling of the points. Put differently, such functions are required

to be invariant to the action of $G = \mathbb{R}^3 \rtimes \text{SO}(3) \times S_n$ on $\mathbb{R}^{3 \times n}$ via

$$\rho_G(t, R, P)(X) = R(X - t1_n^T)P^T, \tag{1}$$

where $t \in \mathbb{R}^3$ defines a translation, $R$ is a rotation and $P$ is a permutation matrix.

Equivariant functions are generalizations of invariant functions: If $G$ acts on $W_1$ via some action $\rho_1(g)$, and on $W_2$ via some other group action $\rho_2(g)$, we say that a function $f : W_1 \to W_2$ is *equivariant* if

$$f(\rho_1(g)w) = \rho_2(g)f(w), \forall w \in W_1 \text{ and } g \in G.$$

Invariant functions correspond to the special case where $\rho_2(g)$ is the identity mapping for all $g \in G$.

In some machine learning tasks on point clouds, the functions learned are not invariant but rather equivariant. For example, segmentation tasks assign a discrete label to each point. They are invariant to translations and rotations but equivariant to permutations – in the sense that permuting the input causes a corresponding permutation of the output. Another example is predicting a normal for each point of a point cloud. This task is invariant to translations but equivariant to both rotations and permutations.

In this paper, we are interested in learning equivariant functions from point clouds into $W_T^n$, where $W_T$ is some representation of $\text{SO}(3)$. The equivariance of these functions is with respect to the action $\rho_G$ on point clouds defined in equation 1, and the action of $G$ on $W_T^n$ defined by applying the rotation action from the left and permutation action from the right as in 1, but 'ignoring' the translation component. Thus, $G$-equivariant functions will be translation invariant. This formulation of equivariance includes the normals prediction example by taking $W_T = \mathbb{R}^3$, as well as the segmentation case by setting $W_T = \mathbb{R}$ with the trivial identity representation. We focus on the harder case of functions into $W_T^n$ which are equivariant to permutations, since it easily implies the easier case of permutation invariant functions to $W_T$.

**Notation.** We use the notation $\mathbb{N}_+ = \mathbb{N} \cup \{0\}$ and $\mathbb{N}_+^* = \bigcup_{r \in \mathbb{N}} \mathbb{N}_+^r$. We set $[D] = \{1, \dots, D\}$ and $[D]_0 = \{0, \dots, D\}$.

**Proofs.** Proofs appear in the appendices, arranged according to sections.

## 3.2 CONDITIONS FOR UNIVERSALITY

**The semi-lifted approach** In general, highly expressive equivariant neural networks can be achieved by using a 'lifted approach', where intermediate features in the network belong to high dimensional representations of the group. In the context of point clouds where typically $n \gg 3$, many papers, e.g., (Thomas et al., 2018; Kondor, 2018; Bogatskiy et al., 2020) use a 'semi-lifted' approach, where hidden layers hold only higher dimensional representations of $\text{SO}(3)$, but not high order permutation representations. In this subsection, we propose a strategy for achieving universality with the semi-lifted approach.

We begin by an axiomatic formulation of the semi-lifted approach (see illustration in inset): we assume that our neural networks are composed of two main components: the first component is a family $\mathcal{F}_{\text{feat}}$ of parametric continuous $G$-equivariant functions $f_{\text{feat}}$ which map the original point cloud $\mathbb{R}^{3 \times n}$ to a semi-lifted point cloud $W_{\text{feat}}^n = \bigoplus_{i=1}^n W_{\text{feat}}$, where $W_{\text{feat}}$ is a lifted (i.e., high-order) representation of $\text{SO}(3)$.

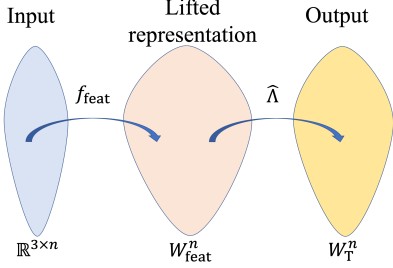

The second component is a family of parametric linear $\text{SO}(3)$-equivariant functions $\mathcal{F}_{\text{pool}}$, which map from the high order representation $W_{\text{feat}}$ down to the target representation $W_T$. Each such $\text{SO}(3)$–equivariant function $\Lambda : W_{\text{feat}} \to W_T$ can be extended to a $\text{SO}(3) \times S_n$ equivariant function $\hat{\Lambda} : W_{\text{feat}}^n \to W_T^n$ by applying $\Lambda$ elementwise. For every positive integer $C$, these two families of functions induce a family of functions $\mathcal{F}_C$ obtained by summing $C$ different compositions of these functions:

$$\mathcal{F}_C(\mathcal{F}_{\text{feat}}, \mathcal{F}_{\text{pool}}) = \left\{ f \,\middle|\, f(X) = \sum_{c=1}^C \hat{\Lambda}_c(g_c(X)), \ (\Lambda_c, g_c) \in \mathcal{F}_{\text{pool}} \times \mathcal{F}_{\text{feat}} \right\}. \tag{2}$$

**Conditions for universality**    We now describe two conditions that guarantee universality using the semi-lifted approach. The first step is showing, as in (Yarotsky, 2018), that continuous $G$-equivariant functions $\mathcal{C}_G(\mathbb{R}^{3 \times n}, W_T^n)$ can be approximated by $G$-equivariant polynomials $\mathcal{P}_G(\mathbb{R}^{3 \times n}, W_T^n)$.

**Lemma 1.** *Any continuous $G$-equivariant function in $\mathcal{C}_G(\mathbb{R}^{3 \times n}, W_T^n)$ can be approximated uniformly on compact sets by $G$-equivariant polynomials in $\mathcal{P}_G(\mathbb{R}^{3 \times n}, W_T^n)$.*

Universality is now reduced to the approximation of $G$-equivariant polynomials. Next, we provide two conditions which guarantee that $G$-equivariant polynomials of degree $D$ can be expressed by function spaces $\mathcal{F}_C(\mathcal{F}_{\text{feat}}, \mathcal{F}_{\text{pool}})$ as defined in equation 2. The idea behind these conditions is that explicit characterizations of polynomials equivariant to the joint action of translations, rotations and permutations is challenging. However, it is possible to explicitly characterize polynomials equivariant to translations and permutations (but not rotations). The key observation is then that this characterization can be rewritten as a sum of functions to $W_{\text{feat}}^n$, a high dimensional representations of SO(3) which is equivariant to translations, permutations *and rotations*, composed with a linear map which is permutation equivariant (but does not respect rotations). Accordingly, our first condition is that $\mathcal{F}_{\text{feat}}$ contains a spanning set of such functions to $W_{\text{feat}}^n$. We call this condition $D$-spanning:

**Definition 1** ($D$-spanning). *For $D \in \mathbb{N}_+$, let $\mathcal{F}_{\text{feat}}$ be a subset of $\mathcal{C}_G(\mathbb{R}^{3 \times n}, W_{\text{feat}}^n)$. We say that $\mathcal{F}_{\text{feat}}$ is $D$-spanning, if there exist $f_1, \ldots, f_K \in \mathcal{F}_{\text{feat}}$, such that every polynomial $p : \mathbb{R}^{3 \times n} \to \mathbb{R}^n$ of degree $D$ which is invariant to translations and equivariant to permutations, can be written as*

$$p(X) = \sum_{k=1}^{K} \hat{\Lambda}_k(f_k(X)), \tag{3}$$

*where $\Lambda_k : W_{\text{feat}} \to \mathbb{R}$ are all linear functionals, and $\hat{\Lambda}_k : W_{\text{feat}}^n \to \mathbb{R}^n$ are the functions defined by elementwise applications of $\Lambda_k$.*

In Lemma 4 we explicitly construct a $D$-spanning family of functions. This provides us with a concrete condition which implies $D$-spanning for other function families as well.

The second condition is that $\mathcal{F}_{\text{pool}}$ contains all SO(3) linear equivariant layers. We call this condition *Linear universality*. Intuitively, taking linear rotation equivariant $\Lambda_k$ in equation 3 ensures that the resulting function $p$ will be rotation equivariant and thus fully $G$-equivariant, and linear universality guarantees the ability to express all such $G$ invariant functions.

**Definition 2** (Linear universality). *We say that a collection $\mathcal{F}_{\text{pool}}$ of equivariant linear functionals between two representations $W_{\text{feat}}$ and $W_T$ of SO(3) is linearly universal, if it contains all linear SO(3)-equivariant mappings between the two representations.*

When these two conditions apply, a rather simple symmetrization arguments leads to the following theorem:

**Theorem 1.** *If $\mathcal{F}_{\text{feat}}$ is $D$-spanning and $\mathcal{F}_{\text{pool}}$ is linearly universal, then there exists some $C(D) \in \mathbb{N}$ such that for all $C \geq C(D)$ the function space $\mathcal{F}_C(\mathcal{F}_{\text{feat}}, \mathcal{F}_{\text{pool}})$ contains all $G$-equivariant polynomials of degree $\leq D$.*

*Proof idea.* By the $D$-spanning assumption, there exist $f_1, \ldots, f_K \in \mathcal{F}_{\text{feat}}$ such that any vector valued polynomial invariant to translations and equivariant to permutations is of the form

$$p(X) = \sum_{k=1}^{K} \hat{\Lambda}_k(f_k(X)), \tag{4}$$

While by definition this holds for functions $p$ whose image is $\mathbb{R}^n$, this is easily extended to functions to $W_T^n$ as well.

It remains to show that when $p$ is also SO(3)-equivariant, we can choose $\Lambda_k$ to be SO(3) equivariant. This is accomplished by averaging over SO(3). □

As a result of Theorem 1 and Lemma 1 we obtain our universality result (see inset for illustration)

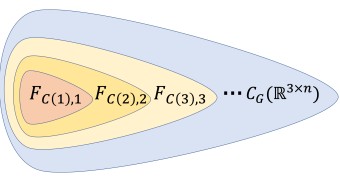

**Corollary 1.** *For all $C, D \in \mathbb{N}_+$, let $\mathcal{F}_{C,D}$ denote function spaces generated by a pair of functions spaces which are $D$-spanning and linearly universal as in equation 2. Then any continuous $G$-equivariant function in* $\mathcal{C}_G(\mathbb{R}^{3 \times n}, W_T^n)$ *can be approximated uniformly on compact sets by equivariant functions in*

$$\mathcal{F} = \bigcup_{D \in \mathbb{N}} \mathcal{F}_{C(D), D}.$$

### 3.3 UNIVERSALITY CONDITIONS IN ACTION

In the remainder of the paper, we prove the universality of several $G$-equivariant architectures, based on the framework we discussed in the previous subsection. We discuss two different strategies for achieving universality, which differ mainly in the type of lifted representations of SO(3) they use: (i) The first strategy uses (direct sums of) tensor-product representations; (ii) The second uses (direct sums of) irreducible representations. The main advantage of the first strategy from the perspective of our methodology is that achieving the $D$-spanning property is more straightforward. The advantage of irreducible representations is that they almost automatically guarantees the linear universality property.

In Section 4 we discuss universality through tensor product representations, and give an example of a minimal tensor representation network architecture that would satisfy universality. In section 5 we discuss universality through irreducible representations, which is currently the more common strategy. We show that the TFN architecture (Thomas et al., 2018; Fuchs et al., 2020) which follows this strategy is universal, and describe a simple tweak that achieves universality using only low order filters, though the representations throughout the network are high dimensional.

## 4 UNIVERSALITY WITH TENSOR REPRESENTATIONS

In this section, we prove universality for models that are based on tensor product representations, as defined below. The main advantage of this approach is that $D$-spanning is achieved rather easily. The main drawbacks are that its data representation is somewhat redundant and that characterizing the linear equivariant layers is more laborious.

**Tensor representations** We begin by defining tensor representations. For $k \in \mathbb{N}_+$ denote $\mathcal{T}_k = \mathbb{R}^{3^k}$. SO(3) acts on $\mathcal{T}_k$ by the tensor product representation, i.e., by applying the matrix Kronecker product $k$ times: $\rho_k(R) := R^{\otimes k}$. The inset illustrates the vector spaces and action for

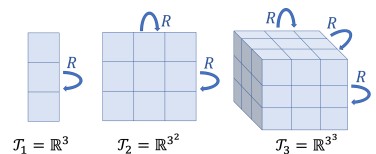

$k = 1, 2, 3$. With this action, for any $i_1, \ldots, i_k \in [n]$, the map from $\mathbb{R}^{3 \times n}$ to $\mathcal{T}_k$ defined by

$$(x_1, \ldots, x_n) \mapsto x_{i_1} \otimes x_{i_2} \ldots \otimes x_{i_k} \tag{5}$$

is SO(3) equivariant.

**A $D$-spanning family** We now show that tensor representations can be used to define a finite set of $D$-spanning functions. The lifted representation $W_{\text{feat}}$ will be given by

$$W_{\text{feat}}^{\mathcal{T}} = \bigoplus_{T=0}^{D} \mathcal{T}_T.$$

The $D$-spanning functions are indexed by vectors $\vec{r} = (r_1, \ldots, r_K)$, where each $r_k$ is a non-negative integer. Denoting $T = \|r\|_1$, the functions $Q^{(\vec{r})} : \mathbb{R}^{3 \times n} \to \mathcal{T}_T^n$, $Q^{(\vec{r})} = (Q_j^{(\vec{r})})_{j=1}^n$ are defined for fixed $j \in [n]$ by

$$Q_j^{(\vec{r})}(X) = \sum_{i_2, \ldots, i_K = 1}^n x_j^{\otimes r_1} \otimes x_{i_2}^{\otimes r_2} \otimes x_{i_3}^{\otimes r_3} \otimes \ldots \otimes x_{i_K}^{\otimes r_K}. \tag{6}$$

The functions $Q_j^{(\vec{r})}$ are $SO(3)$ equivariant as they are a sum of equivariant functions from equation 5. Thus $Q^{(\vec{r})}$ is $SO(3) \times S_n$ equivariant. The motivation behind the definition of these functions is that known characterizations of permutation equivariant polynomials Segol & Lipman (2019) tell us that the entries of these tensor valued functions span all permutation equivariant polynomials (see the proof of Lemma 2 for more details).

To account for translation invariance, we compose the functions $Q^{(\vec{r})}$ with the centralization operation and define the set of functions

$$\mathcal{Q}_D = \{\iota \circ Q^{(\vec{r})}(X - \frac{1}{n}X1_n1_n^T)| \; \|\vec{r}\|_1 \leq D\}, \tag{7}$$

where $\iota$ is the natural embedding that takes each $\mathcal{T}_T$ into $W_{\text{feat}}^{\mathcal{T}} = \bigoplus_{T=0}^{D} \mathcal{T}_T$. In the following lemma, we prove that this set is $D$-spanning.

**Lemma 2.** *For every $D \in \mathbb{N}_+$, the set $\mathcal{Q}_D$ is $D$-spanning.*

*Proof idea.* It is known (Segol & Lipman, 2019) (Theorem 2) that polynomials $p : \mathbb{R}^{3 \times n} \to \mathbb{R}^n$ which are $S_n$-equivariant, are spanned by polynomials of the form $p_{\vec{\alpha}} = (p_{\vec{\alpha}}^j)_{j=1}^n$, defined as

$$p_{\vec{\alpha}}^j(X) = \sum_{i_2,\ldots,i_K=1}^{n} x_j^{\alpha_1} x_{i_2}^{\alpha_2} \ldots x_{i_k}^{\alpha_k} \tag{8}$$

where $\vec{\alpha} = (\alpha_1, \ldots, \alpha_K)$ and each $\alpha_k \in \mathbb{N}_+^3$ is a multi-index. We first show that these polynomials can be extracted from $Q^{(\vec{r})}$ and then use them to represent $p$. $\quad\square$

**A minimal universal architecture** Once we have shown that $\mathcal{Q}_D$ is $D$-spanning, we can design $D$-spanning architectures, by devising architectures that are able to span all elements of $\mathcal{Q}_D$. As we will now show, the compositional nature of neural networks allows us to do this in a very clean manner.

We define a parametric function $f(X, V|\theta_1, \theta_2)$ which maps $\mathbb{R}^{3 \times n} \oplus \mathcal{T}_k^n$ to $\mathbb{R}^{3 \times n} \oplus \mathcal{T}_{k+1}^n$ as follows: For all $j \in [n]$, we have $f_j(X, V) = (x_j, \tilde{V}_j(X, V))$, where

$$\tilde{V}_j(X, V|\theta_1, \theta_2) = \theta_1 \left( x_j \otimes V_j \right) + \theta_2 \sum_i \left( x_i \otimes V_i \right) \tag{9}$$

We denote the set of functions $(X, V) \mapsto f(X, V|\theta_1, \theta_2)$ obtained by choosing the parameters $\theta_1, \theta_2 \in \mathbb{R}$, by $\mathcal{F}_{min}$. While in the hidden layers of our network the data is represented using both coordinates $(X, V)$, the input to the network only contains an $X$ coordinate and the output only contains a $V$ coordinate. To this end, we define the functions

$$\text{ext}(X) = (X, 1_n) \text{ and } \pi_V(X, V) = V. \tag{10}$$

We can achieve $D$-spanning by composition of functions in $\mathcal{F}_{min}$ with these functions and centralizing:

**Lemma 3.** *The function set $\mathcal{Q}_D$ is contained in*

$$\mathcal{F}_{\text{feat}} = \{\iota \circ \pi_V \circ f^1 \circ f^2 \circ \ldots \circ f^T \circ \text{ext}(X - \frac{1}{n}X1_n1_n^T)| \; f^j \in \mathcal{F}_{min}, T \leq D\}. \tag{11}$$

*Thus $\mathcal{F}_{\text{feat}}$ is $D$-spanning.*

*Proof idea.* The proof is technical and follows by induction on $D$. $\quad\square$

To complete the construction of a universal network, we now need to characterize all linear equivariant functions from $W_{\text{feat}}^{\mathcal{T}}$ to the target representation $W_T$. In Appendix G we show how this can be done for the trivial representation $W_T = \mathbb{R}$. This characterization gives us a set of linear functions $\mathcal{F}_{\text{pool}}$, which combined with $\mathcal{F}_{\text{feat}}$ defined in equation 11 (corresponds to $SO(3)$ invariant functions)

gives us a universal architecture as in Theorem 1. However, the disadvantage of this approach is that implementation of the linear functions in $\mathcal{F}_{\text{pool}}$ is somewhat cumbersome.

In the next section we discuss irreducible representations, which give us a systematic way to address linear equivariant mappings into *any* $W_T$. Proving $D$-spanning for these networks is accomplished via the $D$-spanning property of tensor representations, through the following lemma

**Lemma 4.** *If all functions in $\mathcal{Q}_D$ can be written as*

$$\iota \circ Q^{(\vec{r})}(X - \frac{1}{n}X1_n1_n^T) = \sum_{k=1}^{K} \hat{A}_k f_k(X),$$

*where $f_k \in \mathcal{F}_{\text{feat}}$, $A_k : W_{\text{feat}} \to W_{\text{feat}}^{\mathcal{T}}$ and $\hat{A}_k : W_{\text{feat}}^n \to (W_{\text{feat}}^{\mathcal{T}})^n$ is defined by elementwise application of $A_k$, then $\mathcal{F}_{\text{feat}}$ is $D$-spanning.*

We note that as before, $A_k$ are not necessarily $SO(3)$- equivariant.

*Proof idea.* The lemma follows directly from the assumptions. $\qquad\square$

## 5 UNIVERSALITY WITH IRREDUCIBLE REPRESENTATIONS

In this section, we discuss how to achieve universality when using irreducible representations of $SO(3)$. We will begin by defining irreducible representations, and explaining how linear universality is easily achieved by them, while the $D$-spanning properties of tensor representations can be preserved. This discussion can be seen as an interpretation of the choices made in the construction of TFN and similar networks in the literature. We then show that these architectures are indeed universal.

### 5.1 IRREDUCIBLE REPRESENTATIONS OF $SO(3)$

In general, any finite-dimensional representation $W$ of a compact group $H$ can be decomposed into irreducible representations: a subspace $W_0 \subset W$ is $H$-invariant if $hw \in W_0$ for all $h \in H, w \in W_0$. A representation $W$ is irreducible if it has no non-trivial invariant subspaces. In the case of $SO(3)$, all irreducible real representations are defined by matrices $D^{(\ell)}(R)$, called the real Wigner D-matrices, acting on $W_\ell := \mathbb{R}^{2\ell+1}$ by matrix multiplication. In particular, the representation for $\ell = 0, 1$ are $D^{(0)}(R) = 1$ and $D^{(1)}(R) = R$.

**Linear maps between irreducible representations** As mentioned above, one of the main advantages of using irreducible representations is that there is a very simple characterization of all linear equivariant maps between two direct sums of irreducible representations. We use the notation $W_{\boldsymbol{l}}$ for direct sums of irreducible representations, where $\boldsymbol{l} = (\ell_1, \ldots, \ell_K) \in \mathbb{N}_+^K$ and $W_{\boldsymbol{l}} = \bigoplus_{k=1}^K W_{\ell_k}$.

**Lemma 5.** *Let $\boldsymbol{l}^{(1)} = (\ell_1^{(1)}, \ldots, \ell_{K_1}^{(1)})$ and $\boldsymbol{l}^{(2)} = (\ell_1^{(2)}, \ldots, \ell_{K_2}^{(2)})$. A function $\Lambda = (\Lambda_1, \ldots, \Lambda_{K_2})$ is a linear equivariant mapping between $W_{\boldsymbol{l}^{(1)}}$ and $W_{\boldsymbol{l}^{(2)}}$, if and only if there exists a $K_1 \times K_2$ matrix $M$ with $M_{ij} = 0$ whenever $\ell_i^{(1)} \neq \ell_j^{(2)}$, such that*

$$\Lambda_j(V) = \sum_{i=1}^{K_1} M_{ij} V_i \tag{12}$$

*where $V = (V_i)_{i=1}^{K_1}$ and $V_i \in W_{\ell_i^{(1)}}$ for all $i = 1, \ldots, K_1$.*

*Proof idea.* This lemma is a simple generalization of Schur's lemma, a classical tool in representation theory, which asserts that a non-zero linear map between irreducible representations is a scaled multiply of the identity mapping. Lemma 5 was stated in the complex setting in Kondor (2018). While Schur's lemma, and thus Lemma 5, does not always hold for representations over the reals, we observe here that it holds for real irreducible representations of $SO(3)$ since their dimension is always odd. $\qquad\square$

**Clebsch-Gordan decomposition of tensor products** As any finite-dimensional representation of SO(3) can be decomposed into a direct sum of irreducible representations, this is true for tensor representations as well. In particular, the Clebsch-Gordan coefficients provide an explicit formula for decomposing the tensor product of two irreducible representations $W_{\ell_1}$ and $W_{\ell_2}$ into a direct sum of irreducible representations. This decomposition can be easily extended to decompose the tensor product $W_{\boldsymbol{l}_1} \otimes W_{\boldsymbol{l}_2}$ into a direct sum of irreducible representations, where $\boldsymbol{l}_1, \boldsymbol{l}_2$ are now vectors. In matrix notation, this means there is a unitary linear equivariant $U(\boldsymbol{l}_1, \boldsymbol{l}_2)$ mapping of $W_{\boldsymbol{l}_1} \otimes W_{\boldsymbol{l}_2}$ onto $W_{\boldsymbol{l}}$, where the explicit values of $\boldsymbol{l} = \boldsymbol{l}(\boldsymbol{l}_1, \boldsymbol{l}_2)$ and the matrix $U(\boldsymbol{l}_1, \boldsymbol{l}_2)$ can be inferred directly from the case where $\ell_1$ and $\ell_2$ are scalars.

By repeatedly taking tensor products and applying Clebsch-Gordan decompositions to the result, TFN and similar architectures can achieve the $D$-spanning property in a manner analogous to tensor representations, and also enjoy linear universality since they maintain irreducible representations throughout the network.

## 5.2 TENSOR FIELD NETWORKS

We now describe the basic layers of the TFN architecture (Thomas et al., 2018), which are based on irreducible representations, and suggest an architecture based on these layers which can approximate $G$-equivariant maps into *any* representation $W_{\boldsymbol{l}_T}^n, \boldsymbol{l}_T \in \mathbb{N}_+^*$. There are some superficial differences between our description of TFN and the description in the original paper, for more details see Appendix F.

We note that the universality of TFN also implies the universality of Fuchs et al. (2020), which is a generalization of TFN that enables adding an attention mechanism. Assuming the attention mechanism is not restricted to local neighborhoods, this method is at least as expressive as TFN.

TFNs are composed of three types of layers: (i) Convolution (ii) Self-interaction and (iii) Non-linearities. In our architecture, we only use the first two layers types, which we will now describe:[1].

**Convolution.** Convolutional layers involve taking tensor products of a filter and a feature vector to create a new feature vector, and then decomposing into irreducible representations. Unlike in standard CNN, a filter here depends on the input, and is a function $F : \mathbb{R}^3 \to W_{\boldsymbol{l}_D}$, where $\boldsymbol{l}_D = [0, 1, \ldots, D]^T$. The $\ell$-th component of the filter $F(x) = \left[F^{(0)}(x), \ldots, F^{(D)}(x)\right]$ will be given by

$$F_m^{(\ell)}(x) = R^{(\ell)}(\|x\|)Y_m^\ell(\hat{x}), \ m = -\ell, \ldots, \ell \tag{13}$$

where $\hat{x} = x/\|x\|$ if $x \neq 0$ and $\hat{x} = 0$ otherwise, $Y_m^\ell$ are spherical harmonics, and $R^{(\ell)}$ any polynomial of degree $\leq D$. In Appendix F we show that these polynomial functions can be replaced by fully connected networks, since the latter can approximate all polynomials uniformly.

The convolution of an input feature $V \in W_{\boldsymbol{l}_i}^n$ and a filter $F$ as defined above, will give an output feature $\tilde{V} = (\tilde{V}_a)_{a=1}^n \in W_{\boldsymbol{l}_0}^n$, where $\boldsymbol{l}_o = \boldsymbol{l}(\boldsymbol{l}_f, \boldsymbol{l}_i)$, which is given by

$$\tilde{V}_a(X, V) = U(\boldsymbol{l}_f, \boldsymbol{l}_i) \left( \theta_0 V_a + \sum_{b=1}^n F(x_a - x_b) \otimes V_b \right). \tag{14}$$

More formally we will think of convolutional layer as functions of the form $f(X, V) = (X, \tilde{V}(X, V))$. These functions are defined by a choice of $D$, a choice of a scalar polynomial $R^{(\ell)}, \ell = 0, \ldots, D$, and a choice of the parameter $\theta_0 \in \mathbb{R}$ in equation 14. We denote the set of all such functions $f$ by $\mathcal{F}_D$.

**Self Interaction layers.** Self interaction layers are linear functions from $\hat{\Lambda} : W_{\boldsymbol{l}}^n \to W_{\boldsymbol{l}_T}^n$, which are obtained from elementwise application of equivariant linear functions $\Lambda : W_{\boldsymbol{l}} \to W_{\boldsymbol{l}_T}$. These linear functions can be specified by a choice of matrix $M$ with the sparsity pattern described in Lemma 5.

**Activation functions.** TFN, as well as other papers, proposed several activation functions. We find that these layers are not necessary for universality and thus we do not define them here.

**Network architecture.** For our universality proof, we suggest a simple architecture which depends on two positive integer parameters $(C, D)$: For given $D$, we will define $\mathcal{F}_{\text{feat}}(D)$ as the set of

---

[1]Since convolution layers in TFN are not linear, the non-linearities are formally redundant

function obtained by $2D$ recursive convolutions

$$\mathcal{F}_{\text{feat}}(D) = \{\pi_V \circ f^{2D} \circ \ldots f^2 \circ f^1 \circ \text{ext}(X)|\ f^j \in \mathcal{F}_D\},$$

where $\text{ext}$ and $\pi_V$ are defined as in equation 10. The output of a function in $\mathcal{F}_{\text{feat}}(D)$ is in $W_{\boldsymbol{l}(D)}^n$, for some $\boldsymbol{l}(D)$ which depends on $D$. We then define $\mathcal{F}_{\text{pool}}(D)$ to be the self-interaction layers which map $W_{\boldsymbol{l}(D)}^n$ to $W_{\boldsymbol{l}_T}^n$. This choice of $\mathcal{F}_{\text{feat}}(D)$ and $\mathcal{F}_{\text{pool}}(D)$, together with a choice of the number of channels $C$, defines the final network architecture $\mathcal{F}_{C,D}^{\text{TFN}} = \mathcal{F}_C(\mathcal{F}_{\text{feat}}(D), \mathcal{F}_{\text{pool}}(D))$ as in equation 2. In the appendix we prove the universality of TFN:

**Theorem 2.** *For all $n \in \mathbb{N}, \boldsymbol{l}_T \in \mathbb{N}_+^*$,*

1. *For $D \in \mathbb{N}_+$, every $G$-equivariant polynomial $p : \mathbb{R}^{3 \times n} \to W_T^n$ of degree $D$ is in $\mathcal{F}_{C(D),D}^{\text{TFN}}$.*

2. *Every continuous $G$-equivariant function can be approximated uniformly on compact sets by functions in $\cup_{D \in \mathbb{N}_+} \mathcal{F}_{C(D),D}^{\text{TFN}}$*

As discussed previously, the linear universality of $\mathcal{F}_{\text{pool}}$ is guaranteed. Thus proving Theorem 2 amounts to showing that $\mathcal{F}_{\text{feat}}(D)$ is $D$-spanning. This is done using the sufficient condition for $D$-spanning defined in Lemma 4.

*Proof idea.* The proof is rather technical and involved. A useful observation (see Dai & Xu (2013)) used in the proof is that the filters of orders $\ell = 0, 1, \ldots, D$, defined in equation 13, span all polynomial functions of degree $D$ on $\mathbb{R}^3$. This observation is used to show that all functions in $\mathcal{Q}_D$ can be expressed by $\mathcal{F}_{\text{feat}}(D)$ and so $\mathcal{F}_{\text{feat}}$ is $D$-spanning, as stated in Lemma 2. $\qquad\square$

**Alternative architecture** The complexity of the TFN network used to construct $G$-equivariant polynomials of degree $D$, can be reduced using a simple modifications of the convolutional layer in equation 14: We add two parameters $\theta_1, \theta_2 \in \mathbb{R}$ to the convolutional layer, which is now defined as:

$$\tilde{V}_a(X, V) = U(\boldsymbol{l}_f, \boldsymbol{l}_i) \left( \theta_1 \sum_{b=1}^n F(x_a - x_b) \otimes V_b + \theta_2 \sum_{b=1}^n F(x_a - x_b) \otimes V_a \right). \qquad (15)$$

With this simple change, we can show that $\mathcal{F}_{\text{feat}}(D)$ is $D$-spanning even if we only take filters of order 0 and 1 throughout the network. This is shown in Appendix E.

## 6 CONCLUSION

In this paper, we have presented a new framework for proving the universality of $G$-equivariant point cloud networks. We used this framework for proving the universality of the TFN model Thomas et al. (2018); Fuchs et al. (2020), and for devising two additional novel simple universal architectures. In the future we hope to extend these simple constructions to operational $G$-equivariant networks with universality guarantees and competitive practical performance.

Our universal architectures do not require activation functions, and use a single self-interaction layer. In Appendix H we present an experiment indicating that the performance of TFN is not significantly altered by these simplifications. Our architectures also require high order representations, and our experiments show that using increasingly high order representations does indeed improve performance. To date, practical TFN implementation included a relatively small amount of layers, and did not use very high order representations. We believe our theoretical results will inspire interest in stable implementation of larger architectures. On the other hand, an interesting open problem is understanding whether universality can be achieved using only low-dimensional representations.

Finally, we believe that the framework we developed here will be useful for proving the universality of other $G$-equivariant models for point cloud networks, and other related equivariant models. We note that large parts of our discussion can be easily generalized to symmetry groups of the form $G = \mathbb{R}^d \rtimes H \times S_n$ acting on $\mathbb{R}^{d \times n}$, where $H$ can be any compact topological group.

**Acknowledgments** The authors would like to thank Fabian B. Fuchs for making code available and Taco Cohen for helpful discussion. N.D. is supported by THEORINET Simons award 814643.

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

## A  NOTATION

We introduce some notation for the proofs in the appendices. We use the shortened notation $\bar{X} = X - \frac{1}{n}X1_n1_n^T$ and denote the columns of $\bar{X}$ by $(\bar{x}_1, \ldots, \bar{x}_n)$. We denote

$$\Sigma_T = \{\vec{r} \in \mathbb{N}_+^* | \, \|\vec{r}\|_1 = T\}$$

## B  PROOFS FOR SECTION 3

### B.1  $G$-EQUIVARIANT POLYNOMIALS ARE DENSE

A first step in proving denseness of $G$-equivariance polynomials, and in the proof used in the next subsection is the following simple lemma, which shows that translation invariance can be dealt with simply by centralizing the point cloud.

In the following, $\rho_{W_T}$ is some representation of $\mathrm{SO}(3)$ on a finite dimensional real vector space $W_T$. this induces an action $\rho_{W_T \times S_n}$ of $\mathrm{SO}(3) \times S_n$ on $W_T^n$ by

$$\rho_{W_T \times S_n}(R, P)(Y) = \rho_{W_T}(R)YP^T$$

This is also the action of $G$ which we consider, $\rho_G = \rho_{W_T \times S_n}$, where we have invariance with respect to the translation coordinate. The action of $G$ on $\mathbb{R}^{3 \times n}$ is defined in equation 1.

**Lemma B.1.** *A function $f : \mathbb{R}^{3 \times n} \to W_T^n$ is $G$-equivariant, if and only if there exists a function $h$ which is equivariant with respect to the action of $\mathrm{SO}(3) \times S_n$ on $\mathbb{R}^{3 \times n}$, and*

$$f(X) = h(X - \frac{1}{n}X1_n1_n^T) \tag{16}$$

*Proof.* Recall that $G$-equivariance means $\mathrm{SO}(3) \times S_n$ equivariance and translation invariance. Thus if $f$ is $G$-equivariant then equation 16 holds with $h = f$.

On the other hand, if $f$ satisfies equation 16 then we claim it is $G$-equivariant. Indeed, for all $(t, R, P) \in \mathbb{R}^d \rtimes \mathrm{SO}(3) \times S_n$ , since $P^T1_n1_n^T = 1_n1_n^T = 1_n1_n^TP^T$,

$$f\left(\rho_G(t, R, P)(X)\right) = f(R(X + t1_n)P^T) = h(R(X + t1_n)P^T - \frac{1}{n}R(X + t1)P^T1_n1_n^T)$$

$$= h(R(X - \frac{1}{n}X1_n1_n^T)P^T) = h\left(\rho_{\mathbb{R}^3 \times S_n}(R, P)(X - \frac{1}{n}X1_n1_n^T)\right)$$

$$= \rho_{W_T \times S_n}(R, P)h\left(X - \frac{1}{n}X1_n1_n^T\right)$$

$$= \rho_G(t, R, P)f(X).$$

$\square$

We now prove denseness of $G$-equivariant polynomials in the space of $G$-invariant continuous functions (Lemma 1).

**Lemma 1.** *Any continuous $G$-equivariant function in $\mathcal{C}_G(\mathbb{R}^{3 \times n}, W_T^n)$ can be approximated uniformly on compact sets by $G$-equivariant polynomials in $\mathcal{P}_G(\mathbb{R}^{3 \times n}, W_T^n)$.*

*Proof of Lemma 1.* Let $K \subseteq \mathbb{R}^{3 \times n}$ be a compact set. We need to show that continuous $G$-equivariant functions can be approximated uniformly in $K$ by $G$-equivariant polynomials. Let $K_0$ denote the compact set which is the image of $K$ under the centralizing map $X \mapsto X - \frac{1}{n}X1_n1_n^T$. By Lemma B.1, it is sufficient to show that every $\mathrm{SO}(3) \times S_n$ equivariant continuous function $f$ can be approximated uniformly on $K_0$ by a sequence of $\mathrm{SO}(3) \times S_n$ equivariant polynomials $p_k$. The argument is now concluded by the following general lemma:

**Lemma B.2.** *Let $G$ be a compact group, Let $\rho_1$ and $\rho_2$ be continuous[2] representations of $G$ on the Euclidean spaces $W_1$ and $W_2$. Let $K \subseteq W_1$ be a compact set. Then every equivariant function $f : W_1 \to W_2$ can be approximated uniformly on $K$ by a sequence of equivariant polynomials $p_k : W_1 \mapsto W_2$.*

Let $\mu$ be the Haar probability measure associated with the compact group $G$. Let $K_1$ denote the compact set obtained as an image of the compact set $G \times K$ under the continuous mapping

$$(g, X) \mapsto \rho_1(g)X.$$

Using the Stone-Weierstrass theorem, let $p_k$ be a sequence of (not necessarily equivariant) polynomials which approximate $f$ uniformly on $K_1$. Every degree $D$ polynomial $p : W_1 \to W_2$ induces a $G$-equivariant function

$$\langle p \rangle(X) = \int_G \rho_2(g^{-1})p(\rho_1(g)X)d\mu(g).$$

This function $\langle p \rangle$ is a degree $D$ polynomial as well: This is because $\langle p \rangle$ can be approximated uniformly on $K_1$ by "Riemann Sums" of the form $\sum_{j=1}^{N} w_j \rho_2(g_j^{-1})p(\rho_1(g_j)X)$ which are degree $D$ polynomials, and because degree $D$ polynomials are closed in $C(K_1)$.

Now for all $X \in K_1$, continuity of the function $g \mapsto \rho_2(g^{-1})$ implies that the operator norm of $\rho_2(g^{-1})$ is bounded uniformly by some constant $N > 0$, and so

$$|\langle p_k \rangle(X) - f(X)| = \left| \int_G \rho_2(g^{-1})p_k(\rho_1(g)X) - \rho_2(g^{-1})f(\rho_1(g)X)d\mu(g) \right|$$

$$= \left| \int_G \rho_2(g^{-1}) \left[ p_k(\rho_1(g)X) - f(\rho_1(g)X) \right] d\mu(g) \right| \leq N\|f - p_k\|_{C(K_1)} \to 0$$

$\square$

### B.2 PROOF OF THEOREM 1

**Theorem 1.** *If $\mathcal{F}_{\text{feat}}$ is $D$-spanning and $\mathcal{F}_{\text{pool}}$ is linearly universal, then there exists some $C(D) \in \mathbb{N}$ such that for all $C \geq C(D)$ the function space $\mathcal{F}_C(\mathcal{F}_{\text{feat}}, \mathcal{F}_{\text{pool}})$ contains all $G$-equivariant polynomials of degree $\leq D$.*

*Proof.* By the $D$-spanning assumption, there exist $f_1, \ldots, f_K \in \mathcal{F}_{\text{feat}}$ such that any vector valued polynomial $p : \mathbb{R}^{3 \times n} \to \mathbb{R}^n$ invariant to translations and equivariant to permutations is of the form

$$p(X) = \sum_{k=1}^{K} \hat{\Lambda}_k(f_k(X)), \tag{17}$$

where $\Lambda_k$ are linear functions to $\mathbb{R}$. If $p$ is a matrix valued polynomial mapping $\mathbb{R}^{3 \times n}$ to $W_T^n = \mathbb{R}^{t \times n}$, which is invariant to translations and equivariant to permutations, then it is of the form $p = (p_{ij})_{i \in [t], j \in [n]}$, and each $p_i = (p_{ij})_{j \in [n]}$ is itself invariant to translations and permutation equivariant. It follows that matrix valued $p$ can also be written in the form equation 17, the only difference being that the image of the linear functions $\Lambda_k$ is now $\mathbb{R}^t$.

Now let $p : \mathbb{R}^{3 \times n} \to W_T^n$ be a $G$-equivariant polynomial of degree $\leq D$. It remains to show that we can choose $\Lambda_k$ to be SO(3) equivariant. We do this by a symmetrization argument: denote the Haar probability measure on SO(3) by $\nu$, and the action of SO(3) on $W_{\text{feat}}$ and $W_T$ by $\rho_1$ and $\rho_2$ respectively Denote $p = (p_j)_{j=1}^n$ and $f_k = (f_k^j)_{j=1}^n$. For every $j = 1, \ldots, n$, we use the SO(3) equivariance of $p_j$ and $f_k^j$ to obtain

$$p_j(X) = \int_{\text{SO(3)}} \rho_2(R^{-1}) \circ p_j(RX)d\nu(R) = \sum_{k=1}^{K} \int_{\text{SO(3)}} \rho_2(R^{-1}) \circ \Lambda_k \circ f_j^k(RX)d\nu(R)$$

$$= \sum_{k=1}^{K} \int_{\text{SO(3)}} \rho_2(R^{-1}) \circ \Lambda_k \left( \rho_1(R) \circ f_k^j(X) \right) d\nu(R) = \sum_{k=1}^{K} \tilde{\Lambda}_k \circ f_k^j(X),$$

---

[2]By this we mean that the maps $(g, X) \mapsto \rho_j(g)X, j = 1, 2$ are jointly continuous

where $\tilde{\Lambda}_k$ stands for the equivariant linear functional from $W_{\text{feat}}$ to $W_T$, defined for $w \in W_{\text{feat}}$ by

$$\tilde{\Lambda}_k(w) = \int_{\text{SO}(3)} \rho_2(R^{-1}) \circ \Lambda_k \left( \rho_1(R)w \right) d\nu(R).$$

Thus we have shown that $p$ is in $\mathcal{F}_C(\mathcal{F}_{\text{feat}}, \mathcal{F}_{\text{pool}})$ for $C = K$, as required. $\qquad \square$

## C  PROOFS FOR SECTION 4

We prove Lemma 2

**Lemma 2.** *For every $D \in \mathbb{N}_+$, the set $\mathcal{Q}_D$ is $D$-spanning.*

*Proof.* It is known (Segol & Lipman, 2019) (Theorem 2) that polynomials $p : \mathbb{R}^{3 \times n} \to \mathbb{R}^n$ which are $S_n$-equivariant, are spanned by polynomials of the form $p_{\vec{\alpha}} = (p_{\vec{\alpha}}^j)_{j=1}^n$, defined as

$$p_{\vec{\alpha}}^j(X) = \sum_{i_2, \ldots, i_K = 1}^n x_j^{\alpha_1} x_{i_2}^{\alpha_2} \ldots x_{i_k}^{\alpha_k} \tag{18}$$

where $\vec{\alpha} = (\alpha_1, \ldots, \alpha_K)$ and each $\alpha_k \in \mathbb{N}_+^3$ is a multi-index. It follows that $S_n$-equivariant polynomials of degree $\leq D$ are spanned by polynomials of the form $p_{\vec{\alpha}}^j$ where $\sum_{k=1}^K |\alpha_k| \leq D$. Denoting $r_k = |\alpha_k|, k = 1, \ldots K$, the sum of all $r_k$ by $T$, and $\vec{r} = (r_k)_{k=1}^K$, we see that there exists a linear functional $\Lambda_{\vec{\alpha}, \vec{r}} : \mathcal{T}_T \to \mathbb{R}$ such that

$$p_{\vec{\alpha}}^j(X) = \Lambda_{\vec{\alpha}, \vec{r}} \circ Q_j^{\vec{r}}(X)$$

where we recall that $Q^{\vec{r}} = \left( Q_j^{(\vec{r})}(X) \right)_{j=1}^n$ is defined in equation 6 as

$$Q_j^{(\vec{r})}(X) = \sum_{i_2, \ldots, i_K = 1}^n x_j^{\otimes r_1} \otimes x_{i_2}^{\otimes r_2} \otimes x_{i_3}^{\otimes r_3} \otimes \ldots \otimes x_{i_K}^{\otimes r_K}.$$

Thus polynomials $p = (p_j)_{j=1}^n$ which are of degree $\leq D$, and are $S_n$ equivariant, can be written as

$$p_j(X) = \sum_{T \leq D} \sum_{\vec{r} \in \Sigma_T} \sum_{\vec{\alpha} | |\alpha_k| = r_k} \Lambda_{\vec{\alpha}, \vec{r}} \left( Q_j^{(\vec{r})}(X) \right) = \sum_{T \leq D} \sum_{\vec{r} \in \Sigma_T} \Lambda_{\vec{r}} \left( \iota \circ Q_j^{(\vec{r})}(X) \right), j = 1, \ldots, n,$$

where $\Lambda_{\vec{r}} = \sum_{\vec{\alpha} | |\alpha_k| = r_k} \Lambda_{\vec{\alpha}, \vec{r}} \circ \iota_T^{-1}$, and $\iota_T^{-1}$ is the left inverse of the embedding $\iota$. If $p$ is also translation invariant, then

$$p(X) = p(X - \frac{1}{n}X 1_n 1_n^T) = \sum_{T \leq D} \sum_{\vec{r} \in \Sigma_T} \hat{\Lambda}_{\vec{r}} \left( \iota \circ Q^{(\vec{r})}(X - \frac{1}{n}X 1_n 1_n^T) \right).$$

Thus $\mathcal{Q}_D$ is $D$-spanning. $\qquad \square$

We prove Lemma 3

**Lemma 3.** *The function set $\mathcal{Q}_D$ is contained in*

$$\mathcal{F}_{\text{feat}} = \{\iota \circ \pi_V \circ f^1 \circ f^2 \circ \ldots \circ f^T \circ \text{ext}(X - \frac{1}{n}X 1_n 1_n^T) | f^j \in \mathcal{F}_{min}, T \leq D\}. \tag{11}$$

*Thus $\mathcal{F}_{\text{feat}}$ is $D$-spanning.*

*Proof.* In this proof we make the dependence of $\mathcal{F}_{\text{feat}}$ on $D$ explicit and denote $\mathcal{F}_{\text{feat}}(D)$.

We prove the claim by induction on $D$. Assume $D = 0$. Then $\mathcal{Q}_0$ contains only the constant function $X \mapsto 1_n \in \mathcal{T}_0^n$, and this is precisely the function $\pi_V \circ \text{ext} \in \mathcal{F}_{\text{feat}}(0)$.

Now assume the claim holds for all $D'$ with $D - 1 \geq D' \geq 0$, and prove the claim for $D$. Choose $\vec{r} = (r_1, \ldots, r_k) \in \Sigma_T$ for some $T \leq D$, we need to show that the function $Q^{(\vec{r})}$ is in $\mathcal{F}_{\text{feat}}(D)$. Since $\mathcal{F}_{\text{feat}}(D - 1) \subseteq \mathcal{F}_{\text{feat}}(D)$ we know from the induction hypothesis that this is true if $T < D$. Now assume $T = D$. We consider two cases:

1. If $r_1 > 0$, we set $\tilde{r} = (r_1 - 1, r_2, \ldots, r_K)$. We know that $\iota \circ Q^{(\tilde{r})}(\bar{X}) \in \mathcal{F}_{\text{feat}}(D-1)$ by the induction hypothesis. So there exist $f_2, \ldots, f_D$ such that

$$\iota \circ \pi_V \circ f_2 \circ \ldots \circ f_D \circ \text{ext}(\bar{X}) = \iota \circ Q^{(\tilde{r})}(\bar{X}). \tag{19}$$

Now choose $f_1 \in \mathcal{F}_{min}$ to be the function whose $V$ coordinate $\tilde{V} = (\tilde{V}_j)_{j=1}^n$, is given by $\tilde{V}_j(X, V) = x_j \otimes V_j$, obtained by setting $\theta_1 = 1, \theta_2 = 0$ in equation 9. Then , we have

$$\tilde{V}_j(\bar{X}, Q^{(\tilde{r})}(\bar{X})) = \sum_{i_2,\ldots,i_K=1}^n \bar{x}_j \otimes \bar{x}_j^{\otimes(r_1-1)} \otimes \bar{x}_{i_2}^{\otimes r_2} \otimes \ldots \otimes \bar{x}_{i_K}^{\otimes r_K}$$
$$= Q_j^{(\vec{r})}(\bar{X}).$$

and so

$$\iota \circ \pi_V \circ f_1 \circ f_2 \circ \ldots \circ f_D \circ \text{ext}(X - \frac{1}{n} X 1_n 1_n^T) = \iota \circ Q^{(\vec{r})}(\bar{X}). \tag{20}$$

and $\iota \circ Q^{(\vec{r})}(X - \frac{1}{n} X 1_n 1_n^T) \in \mathcal{F}_{\text{feat}}(D)$.

2. If $r_1 = 0$. We assume without loss of generality that $r_2 > 0$. Set $\tilde{r} = (r_2 - 1, r_3, \ldots, r_K)$. As before by the induction hypothesis there exist $f_2, \ldots, f_D$ which satisfy equation 19. This time we choose $f_1 \in \mathcal{F}_{min}$ to be the function whose $V$ coordinate $\tilde{V} = (\tilde{V}_j)_{j=1}^n$, is given by $\tilde{V}_j(X, V) = \sum_j x_j \otimes V_j$, obtained by setting $\theta_1 = 0, \theta_2 = 1$ in equation 9. Then we have

$$\tilde{V}_j(\bar{X}, Q^{(\tilde{r})}(\bar{X})) = \sum_{j=1}^n \sum_{i_3,\ldots,i_K=1}^n \bar{x}_j \otimes \bar{x}_j^{\otimes(r_2-1)} \otimes \bar{x}_{i_3}^{\otimes r_2} \otimes \ldots \otimes \bar{x}_{i_K}^{\otimes r_K}$$
$$= \sum_{i_2,i_3,\ldots,i_K=1}^n \bar{x}_{i_2}^{\otimes r_2} \otimes \bar{x}_{i_3}^{\otimes r_2} \otimes \ldots \otimes \bar{x}_{i_K}^{\otimes r_K}$$
$$= Q_j^{(\vec{r})}(\bar{X}).$$

Thus equation 20 holds, and so again we have that $\iota \circ Q^{(\vec{r})}(X - \frac{1}{n} X 1_n 1_n^T) \in \mathcal{F}_{\text{feat}}(D)$.

$\square$

Finally we prove Lemma 4

**Lemma 4.** *If all functions in $\mathcal{Q}_D$ can be written as*

$$\iota \circ Q^{(\vec{r})}(X - \frac{1}{n} X 1_n 1_n^T) = \sum_{k=1}^K \hat{A}_k f_k(X),$$

*where $f_k \in \mathcal{F}_{\text{feat}}$, $A_k : W_{\text{feat}} \to W_{\text{feat}}^{\mathcal{T}}$ and $\hat{A}_k : W_{\text{feat}}^n \to (W_{\text{feat}}^{\mathcal{T}})^n$ is defined by elementwise application of $A_k$, then $\mathcal{F}_{\text{feat}}$ is $D$-spanning.*

*Proof.* If the conditions in Lemma 4 hold, then since $\mathcal{Q}_D$ is $D$-spanning, every translation invariant and permutation equivariant polynomials $p$ of degree $D$ can be written as

$$p(X) = \sum_{\vec{r} \mid \|\vec{r}\|_1 \leq D} \hat{\Lambda}_{\vec{r}} \left( \iota \circ Q^{(\vec{r})}(X - \frac{1}{n} X 1_n 1_n^T) \right) = \sum_{\vec{r} \mid \|\vec{r}\|_1 \leq D} \hat{\Lambda}_{\vec{r}} \left( \sum_{k=1}^{K_{\vec{r}}} \iota \circ \hat{A}_{k,\vec{r}} f_{k,\vec{r}}(X) \right)$$
$$= \sum_{\vec{r} \mid \|\vec{r}\|_1 \leq D} \sum_{k=1}^{K_{\vec{r}}} \hat{\Lambda}_{k,\vec{r}} (f_{k,\vec{r}}(X))$$

where we denote $\Lambda_{k,\vec{r}} = \Lambda_{\vec{r}} \circ \iota \circ A_{k,\vec{r}}$. Thus we proved $\mathcal{F}_{\text{feat}}$ is $D$-spanning. $\square$

## D  PROOFS FOR SECTION 5

We prove Lemma 5

**Lemma 5.** *Let $\boldsymbol{l}^{(1)} = (\ell_1^{(1)}, \ldots, \ell_{K_1}^{(1)})$ and $\boldsymbol{l}^{(2)} = (\ell_1^{(2)}, \ldots, \ell_{K_2}^{(2)})$. A function $\Lambda = (\Lambda_1, \ldots, \Lambda_{K_2})$ is a linear equivariant mapping between $W_{\boldsymbol{l}^{(1)}}$ and $W_{\boldsymbol{l}^{(2)}}$, if and only if there exists a $K_1 \times K_2$ matrix $M$ with $M_{ij} = 0$ whenever $\ell_i^{(1)} \neq \ell_j^{(2)}$, such that*

$$\Lambda_j(V) = \sum_{i=1}^{K_1} M_{ij} V_i \tag{12}$$

*where $V = (V_i)_{i=1}^{K_1}$ and $V_i \in W_{\ell_i^{(1)}}$ for all $i = 1, \ldots, K_1$.*

*Proof.* As mentioned in the main text, this lemma is based on Schur's lemma. This lemma is typically stated for complex representations, but holds for odd dimensional real representation as well. We recount the lemma and its proof here for completeness (see also (Fulton & Harris, 2013)).

**Lemma D.1** (Schur's Lemma for SO(3)). *Let $\Lambda : W_{\ell_1} \to W_{\ell_2}$ be a linear equivariant map. If $\ell_1 \neq \ell_2$ then $\Lambda = 0$. Otherwise $\Lambda$ is a scalar multiply of the identity.*

*Proof.* Let $\Lambda : W_{\ell_1} \to W_{\ell_2}$ be a linear equivariant map. The image and kernel of $\Lambda$ are invariant subspaces of $W_{\ell_1}$ and $W_{\ell_2}$, respectively. It follows that if $\Lambda \neq 0$ then $\Lambda$ is a linear isomorphism so necessarily $\ell_1 = \ell_2$. Now assume $\ell_1 = \ell_2$. Since the dimension of $W_{\ell_1}$ is odd, $\Lambda$ has a real eigenvalue $\lambda$. The linear function $\Lambda - \lambda I$ is equivariant and has a non-trivial kernel, so $\Lambda - \lambda I = 0$. $\square$

We now return to the proof of Lemma 5. Note that each $\Lambda_j : W_{\boldsymbol{l}^{(1)}} \to W_{\ell_j^{(2)}}$ is linear and SO(3) equivariant. Next denote the restrictions of each $\Lambda_j$ to $W_{\ell_i^{(1)}}, i = 1, \ldots, K_2$ by $\Lambda_{ij}$, and note that

$$\Lambda_j(V_1, \ldots, V_{K_1}) = \sum_{i=1}^{K_1} \Lambda_{ij}(V_i). \tag{21}$$

By considering vectors in $W_{\boldsymbol{l}^{(1)}}$ of the form $(0, \ldots, 0, V_i, 0 \ldots, 0)$ we see that each $\Lambda_{ij} : W_{\ell_i^{(1)}} \to W_{\ell_j^{(2)}}$ is linear and SO(3)-equivariant. Thus by Schur's lemma, if $\ell_i^{(1)} = \ell_j^{(2)}$ then $\Lambda_{ij}(V_i) = M_{ij} V_i$ for some real $M_{ij}$, and otherwise $M_{ij} = 0$. Plugging this into equation 21 we obtain equation 12. $\square$

We prove Theorem 2 which shows that the TFN network described in the main text is universal:

**Theorem 2.** *For all $n \in \mathbb{N}, \boldsymbol{l}_T \in \mathbb{N}_+^*$,*

1. *For $D \in \mathbb{N}_+$, every $G$-equivariant polynomial $p : \mathbb{R}^{3 \times n} \to W_T^n$ of degree $D$ is in $\mathcal{F}_{C(D), D}^{\mathrm{TFN}}$.*

2. *Every continuous $G$-equivariant function can be approximated uniformly on compact sets by functions in $\cup_{D \in \mathbb{N}_+} \mathcal{F}_{C(D), D}^{\mathrm{TFN}}$*

*Proof.* As mentioned in the main text, we only need to show that the function space $\mathcal{F}_{\mathrm{feat}}(D)$ is $D$-spanning. Recall that $\mathcal{F}_{\mathrm{feat}}(D)$ is obtained by $2D$ consecutive convolutions with $D$-filters. In general, we denote the space of functions defined by applying $J$ consecutive convolutions by $\mathcal{G}_{J, D}$.

If $\mathcal{Y}$ is a space of functions from $\mathbb{R}^{3 \times n} \to Y^n$, we denote by $\langle \mathcal{Y}, \mathcal{T}_T \rangle$ the space of all functions $p : \mathbb{R}^{3 \times n} \to \mathcal{T}_T^n$ of the form

$$p(X) = \sum_{k=1}^K \hat{A}_k f_k(X), \tag{22}$$

where $A_k : Y \to \mathcal{T}_T$ are linear functions, $\hat{A}_k : Y^n \to \mathcal{T}_T^n$ are induced by elementwise application, and $f_k \in \mathcal{Y}$. This notation is useful because: (i) by Lemma 4 it is sufficient to show that $Q^{(\vec{r})}(\bar{X})$ is in $\langle \mathcal{G}_{2D,D}, \mathcal{T}_T \rangle$ for all $\vec{r} \in \Sigma_T$ and all $T \le D$, and because (ii) it enables comparison of the expressive power of function spaces $\mathcal{Y}_1, \mathcal{Y}_2$ whose elements map to different spaces $Y_1^n, Y_2^n$, since the elements in $\langle \mathcal{Y}_i, \mathcal{T}_T \rangle, i = 1, 2$ both map to the same space. In particular, note that if for every $f \in \mathcal{Y}_2$ there is a $g \in \mathcal{Y}_1$ and a linear map $A : Y_1 \to Y_2$ such that $f(X) = \hat{A} \circ g(X)$, then $\langle \mathcal{Y}_2, \mathcal{T}_T \rangle \subseteq \langle \mathcal{Y}_1, \mathcal{T}_T \rangle$.

We now use this abstract discussion to prove some useful results: the first is that for the purpose of this lemma, we can 'forget about' the multiplication by a unitary matrix in equation 14, used for decomposition into irreducible representations: To see this, denote by $\tilde{\mathcal{G}}_{J,D}$ the function space obtained by taking $J$ consecutive convolutions with $D$-filters without multiplying by a unitary matrix in equation 14. Since Kronecker products of unitary matrices are unitary matrices, we obtain that the elements in $\mathcal{G}_{J,D}$ and $\tilde{\mathcal{G}}_{J,D}$ differ only by multiplication by a unitary matrix, and thus $\langle \tilde{\mathcal{G}}_{J,D}, \mathcal{T}_T \rangle \subseteq \langle \mathcal{G}_{J,D}, \mathcal{T}_T \rangle$ and $\langle \mathcal{G}_{J,D}, \mathcal{T}_T \rangle \subseteq \langle \tilde{\mathcal{G}}_{J,D}, \mathcal{T}_T \rangle$, so both sets are equal.

Next, we prove that adding convolutional layers (enlarging $J$) or taking higher order filters (enlarging $D$) can only increase the expressive power of a network.

**Lemma D.2.** *For all $J, D, T \in \mathbb{N}_+$,*

1. *$\langle \mathcal{G}_{J,D}, \mathcal{T}_T \rangle \subseteq \langle \mathcal{G}_{J+1,D}, \mathcal{T}_T \rangle$.*

2. *$\langle \mathcal{G}_{J,D}, \mathcal{T}_T \rangle \subseteq \langle \mathcal{G}_{J,D+1}, \mathcal{T}_T \rangle$.*

*Proof.* The first claim follows from the fact that every function $f$ in $\langle \mathcal{G}_{J,D}, \mathcal{T}_T \rangle$ can be identified with a function in $\langle \mathcal{G}_{J+1,D}, \mathcal{T}_T \rangle$ by taking the $J + 1$ convolutional layer in equation 14 with $\theta_0 = 1, F = 0$.

The second claim follows from the fact that $D$-filters can be identified with $D + 1$-filters whose $D + 1$-th entry is 0. □

The last preliminary lemma we will need is

**Lemma D.3.** *For every $J, D \in \mathbb{N}_+$, and every $t, s \in \mathbb{N}_+$, if $p \in \langle \mathcal{G}_{J,D}, \mathcal{T}_t \rangle$, then the function $q$ defined by*

$$q_a(X) = \sum_{b=1}^{n} (\bar{x}_a - \bar{x}_b)^{\otimes s} \otimes p_b(X)$$

*is in $\langle \mathcal{G}_{J+1,D}, \mathcal{T}_{t+s} \rangle$.*

*Proof.* This lemma is based on the fact that the space of $s$ homogeneous polynomial on $\mathbb{R}^3$ is spanned by polynomials of the form $\|x\|^{s-\ell} Y_m^\ell(x)$ for $\ell = s, s - 2, s - 4 \dots$ (Dai & Xu, 2013). For each such $\ell$, and $s \le D$, these polynomials can be realized by filters $F^{(\ell)}$ by setting $R^{(\ell)}(\|x\|) = \|x\|^s$ so that

$$F_m^{(\ell)}(x) = \|x\|^s Y_m^\ell(\hat{x}) = \|x\|^{s-\ell} Y_m^\ell(x).$$

For every $D \in \mathbb{N}$ and $s \le D$, we can construct a $D$-filter $F^{s,D} = (F^{(0)}, \dots, F^{(D)})$ where $F^{(s)}, F^{(s-2)}, \dots$ are as defined above and the other filters are zero. Since both the entries of $F^{s,D}(x)$, and the entries of $x^{\otimes s}$, span the space of $s$-homogeneous polynomials on $\mathbb{R}^3$, it follows that there exists a linear mapping $B_s : W_{l_D} \to \mathcal{T}_s$ so that

$$x^{\otimes s} = B_s(F^{s,D}(x)), \forall x \in \mathbb{R}^3. \tag{23}$$

Thus, since $p$ can be written as a sum of compositions of linear mappings with functions in $\mathcal{G}_{J,D}$ as in equation 22, and similarly $x^{\otimes s}$ is obtained as a linear image of functions in $\mathcal{G}_{1,D}$ as in equation 23, we deduce that

$$\sum_{b=1}^{n} (x_a - x_b) \otimes p_b(X) = \sum_{b=1}^{n} (\bar{x}_a - \bar{x}_b) \otimes p_b(X)$$

*is in $\langle \mathcal{G}_{J+1,D}, \mathcal{T}_{t+s} \rangle$* □

As a final preliminary, we note that $D$-filters can perform an averaging operation by setting $R^{(0)} = 1$ and $\theta_0, R^{(1)}, \ldots, R^{(D)} = 0$ in equation 13 and equation 14 . We call this $D$-filter an *averaging filter*.

We are now ready to prove our claim: we need to show that for every $D, T \in \mathbb{N}_+$ where $T \leq D$, for every $\vec{r} \in \Sigma_T$, the function $Q^{(\vec{r})}$ is in $\langle \mathcal{G}_{2D,D}, \mathcal{T}_T \rangle$. Note that due to the inclusion relations in Lemma D.2 it is sufficient to prove this for the case $T = D$. We prove this by induction on $D$. For $D = 0$, vectors $\vec{r} \in \Sigma_0$ contains only zeros and so

$$Q^{(\vec{r})}(\bar{X}) = 1_n = \pi_V \circ \text{ext}(X) \in \langle \mathcal{G}_{0,0}, \mathcal{T}_0 \rangle.$$

We now assume the claim is true for all $D'$ with $D > D' \geq 0$ and prove the claim is true for $D$. We need to show that for every $\vec{r} \in \Sigma_D$ the function $Q^{(\vec{r})}$ is in $\langle \mathcal{G}_{2D,D}, \mathcal{T}_D \rangle$. We prove this yet again by induction, this time on the value of $r_1$: assume that $\vec{r} \in \Sigma_D$ and $r_1 = 0$.. Denote by $\tilde{r}$ the vector in $\Sigma_{D-1}$ defined by

$$\tilde{r} = (r_2 - 1, r_3, \ldots, r_K).$$

By the induction assumption on $D$, we know that $Q^{(\tilde{r})}(\bar{X}) \in \mathcal{G}_{2(D-1),D-1,D-1}$ and so

$$q_a(X) = \sum_{b=1}^{n} (\bar{x}_a - \bar{x}_b) \otimes Q_b^{(\tilde{r})}(\bar{X}) = \sum_{b=1}^{n} (\bar{x}_a - \bar{x}_b) \otimes \bar{x}_b^{\otimes r_2 - 1} \otimes \sum_{i_3, \ldots, i_K = 1}^{n} \bar{x}_{i_3}^{\otimes r_3} \otimes \ldots \otimes \bar{x}_{i_K}^{\otimes r_K}$$

$$= \left( \bar{x}_a \otimes \sum_{b=1}^{n} Q_b^{(\tilde{r})}(\bar{X}) \right) - Q^{(\vec{r})}(\bar{X})$$

is in $\langle \mathcal{G}_{2D-1,D-1}, \mathcal{T}_D \rangle$ by Lemma D.3, which is contained in $\langle \mathcal{G}_{2D-1,D}, \mathcal{T}_D \rangle$ by Lemma D.2. Since $\bar{x}_a$ has zero mean, while $Q_a^{(\vec{r})}(\bar{X})$ does not depend on $a$ since $r_1 = 0$, applying an averaging filter to $q_a$ gives us a constant value $-Q_a^{(\vec{r})}(\bar{X})$ in each coordinate $a \in [n]$, and so $Q^{(\vec{r})}(\bar{X})$ is in $\langle \mathcal{G}_{2D,D}, \mathcal{T}_D \rangle$.

Now assume the claim is true for all $\vec{r} \in \Sigma_D$ which sum to $D$, and whose first coordinate is smaller than some $r_1' \geq 1$, we now prove the claim is true when the first coordinate of $\vec{r}$ is equal to $r_1'$. The vector $\tilde{r} = (r_2, \ldots, r_K)$ obtained from $\vec{r}$ by removing the first coordinate, sums to $D' = D - r_1' < D$, and so by the induction hypothesis on $D$ we know that $Q^{(\tilde{r})} \in \langle \mathcal{G}_{2D',D'}, \mathcal{T}_{D'} \rangle$. By Lemma D.3 we obtain a function $q_a \in \langle \mathcal{G}_{2D'+1,D'}, \mathcal{T}_D \rangle \subseteq \langle \mathcal{G}_{2D,D}, \mathcal{T}_D \rangle$ defined by

$$q_a(X) = \sum_{b=1}^{n} (\bar{x}_a - \bar{x}_b)^{\otimes r_1} \otimes Q_b^{(\tilde{r})}(\bar{X})$$

$$= \sum_{b=1}^{n} (\bar{x}_a - \bar{x}_b)^{\otimes r_1} \otimes \bar{x}_b^{\otimes r_2} \otimes \sum_{i_3, \ldots, i_K = 1}^{n} \bar{x}_{i_3}^{\otimes r_3} \otimes \ldots \otimes \bar{x}_{i_K}^{\otimes r_K}$$

$$= Q_a^{(\vec{r})}(\bar{X}) + \text{additional terms}$$

where the additional terms are linear combinations of functions of the form $P_D Q_a^{(r')}(\bar{X})$ where $r' \in \Sigma_D$ and their first coordinate $r_1$ is smaller than $r_1'$, and $P_D : \mathcal{T}_D \to \mathcal{T}_D$ is a permutation. By the induction hypothesis on $r_1$, each such $Q^{(r')}$ is in $\langle \mathcal{G}_{2D,D}, \mathcal{T}_D \rangle$. It follows that $P_D Q_a^{(r')}(\bar{X}), a = 1, \ldots, n$, and thus $Q^{(\vec{r})}(\bar{X})$, are in $\langle \mathcal{G}_{2D,D}, \mathcal{T}_D \rangle$ as well. This concludes the proof of Theorem 2.

$\square$

# E    ALTERNATIVE TFN ARCHITECTURE

In this appendix we show that replacing the standard TFN convolutional layer with the layer defined in equation 15:

$$\tilde{V}_a(X, V) = U(l_f, l_i) \left( \theta_1 \sum_{b=1}^{n} F(x_a - x_b) \otimes V_b + \theta_2 \sum_{b=1}^{n} F(x_a - x_b) \otimes V_a \right),$$

we can obtain $D$-spanning networks using $2D$ consecutive convolutions with 1-filters (that is, filters in $W_{l_1}$, where $l_1 = [0, 1]^T$). Our discussion here is somewhat informal, meant to provide the general

ideas without delving into the details as we have done for the standard TFN architecture in the proof of Theorem 2. In the end of our discussion we will explain what is necessary to make this argument completely rigorous.

We will only need two fixed filters for our argument here: The first is the 1-filter $F_{Id} = (F^{(0)}, F^{(1)})$ defined by setting $R^{(0)}(\|x\|) = 0$ and $R^{(1)}(\|x\|) = \|x\|$ to obtain

$$F_{Id}(x) = \|x\| Y^1(\hat{x}) = \|x\| \hat{x} = x.$$

The second is the filter $F_1$ defined by setting $R^{(0)}(\|x\|) = 1$ and $R^{(1)}(\|x\|) = 0$, so that

$$F_1(x) = 1.$$

We prove our claim by showing that a pair of convolutions with 1-filters can construct any convolutional layer defined in equation 9 for the $D$-spanning architecture using tensor representations. The claim then follows from the fact that $D$ convolutions of the latter architecture suffice for achieving $D$-spanning, as shown in Lemma 3.

Convolutions for tensor representations, defined in equation 9, are composed of two terms:

$$\tilde{V}_a^{\text{tensor},1}(\bar{X}, V) = \bar{x}_a \otimes V_a \text{ and } \tilde{V}_a^{\text{tensor},2}(\bar{X}, V) = \sum_{b=1}^n \bar{x}_b \otimes V_b.$$

To obtain the first term $\tilde{V}_a^{\text{tensor},1}$, we set $\theta_1 = 0, \theta_2 = 1/n, F = F_{Id}$ in equation 15 we obtain (the decomposition into irreducibles of) $\tilde{V}_a^{\text{tensor},1}(\bar{X}, V) = \bar{x}_a \otimes V_a$. Thus this term can in fact be expressed by a single convolution. We can leave this outcome unchanged by a second convolution, defined by setting $\theta_1 = 0, \theta_2 = 1/n, F = F_1$.

To obtain the second term $\tilde{V}_a^{\text{tensor},2}$, we apply a first convolution with $\theta_1 = -1, F = F_{Id}, \theta_2 = 0$, to obtain

$$\sum_{b=1}^n (x_b - x_a) \otimes V_b = \sum_{b=1}^n (\bar{x}_b - \bar{x}_a) \otimes V_b = \tilde{V}_a^{\text{tensor},2}(V, \bar{X}) - \bar{x}_a \otimes \sum_{b=1}^n V_b$$

By applying an additional averaging filter, defined by setting $\theta_1 = \frac{1}{n}, F = F_1, \theta_2 = 0$, we obtain $\tilde{V}_a^{\text{tensor},2}(V, \bar{X})$. This concludes our 'informal proof'.

Our discussion here has been somewhat inaccurate, since in practice $F_{Id}(x) = (0, x) \in W_0 \oplus W_1$ and $F_1(x) = (1, 0) \in W_0 \oplus W_1$. Moreover, in our proof we have glossed over the multiplication by the unitary matrix used to obtain decomposition into irreducible representations. However the ideas discussed here can be used to show that $2D$ convolutions with 1-filters can satisfy the sufficient condition for $D$-spanning defined in Lemma 4. See our treatment of Theorem 2 for more details.

## F  COMPARISON WITH ORIGINAL TFN PAPER

In this Appendix we discuss three superficial differences between the presentation of the TFN architecture in Thomas et al. (2018) and our presentation here:

1. We define convolutional layers between features residing in direct sums of irreducible representations, while (Thomas et al., 2018) focuses on features which inhabit a single irreducible representation. This difference is non-essential, as direct sums of irreducible representations can be represented as multiple channels where each feature inhabits a single irreducible representation.

2. The term $\theta_0 V_a$ in equation 14 appears in (Fuchs et al., 2020), but does not appear explicitly in (Thomas et al., 2018). However it can be obtained by concatenation of the input of a self-interaction layer to the output, and then applying a self-interaction layer.

3. We take the scalar functions $R^{(\ell)}$ to be polynomials, while (Thomas et al., 2018) take them to be fully connected networks composed with radial basis functions. Using polynomial scalar bases is convenient for our presentation here since it enables exact expression of

equivariant polynomials. Replacing polynomial bases with fully connected networks, we obtain approximation of equivariant polynomials instead of exact expression. It can be shown that if $p$ is a $G$-equivariant polynomial which can be expressed by some network $\mathcal{F}_{C,D}$ defined with filters coming from a polynomial scalar basis, then $p$ can be approximated on a compact set $K$, up to an arbitrary $\epsilon$ error, by a similar network with scalar functions coming from a sufficiently large fully connected network.

## G  TENSOR UNIVERSALITY

In this section we show how to construct the complete set $\mathcal{F}_{\text{pool}}$ of linear SO(3) invariant functionals from $W_{\text{feat}}^{\mathcal{T}} = \bigoplus_{T=0}^{D} \mathcal{T}_T$ to $\mathbb{R}$. Since each such functional $\Lambda$ is of the form

$$\Lambda(w_0, \ldots, w_D) = \sum_{T=0}^{D} \Lambda_T(w_T),$$

where each $\Lambda_T$ is SO(3)-invariant, it is sufficient to characterize all linear SO(3)-invariant functionals $\Lambda : \mathcal{T}_D \to \mathbb{R}$.

It will be convenient to denote

$$W = \mathbb{R}^3 \text{ and } W^{\otimes D} \cong \mathbb{R}^{3^D} = \mathcal{T}_D.$$

We achieve our characterization using the bijective correspondence between linear functional $\Lambda : W^{\otimes D} \to \mathbb{R}$ and multi-linear functions $\tilde{\Lambda} : W^D \to \mathbb{R}$: each such $\Lambda$ corresponds to a unique $\hat{\Lambda}$, such that

$$\tilde{\Lambda}(e_{i_1}, \ldots, e_{i_D}) = \Lambda(e_{i_1} \otimes \ldots \otimes e_{i_D}), \ \forall (i_1, \ldots, i_D) \in [3]^D, \tag{24}$$

where $e_1, e_2, e_3$ denote the standard basis elements of $\mathbb{R}^3$. We define a spanning set of equivariant linear functionals on $W^{\otimes D}$ via a corresponding characterization for multi-linear functionals on $W^D$. Specifically, set

$$K_D = \{k \in \mathbb{N}_+ | D - 3k \text{ is even and non-negative. }\}$$

For $k \in K_D$ we define a multi-linear functional:

$$\tilde{\Lambda}_k(w_1, \ldots, w_D) = \det(w_1, w_2, w_3) \times \ldots \times \det(w_{3k-2}, w_{3k-1}, w_{3k}) \times \langle w_{3k+1}, w_{3k+2} \rangle \times \ldots$$
$$\times \langle w_{D-1}, w_D \rangle, \tag{25}$$

and for $(k, \sigma) \in K_D \times S_D$ we define

$$\tilde{\Lambda}_{k,\sigma}(w_1, \ldots, w_D) = \tilde{\Lambda}_k(w_{\sigma(1)}, \ldots, w_{\sigma(D)}) \tag{26}$$

**Proposition 1.** *The space of linear invariant functions from $\mathcal{T}_D$ to $\mathbb{R}$ is spanned by the set of linear invariant functionals $\lambda_D = \{\Lambda_{k,\sigma} | (k, \sigma) \in K_D \times S_D\}$ induced by the multi-linear functional $\tilde{\Lambda}_{k,\sigma}$ described in equation 25 and equation 26*

We note that (i) equation 24 provides a (cumbersome) way to compute all linear invariant functionals $\Lambda_{k,\sigma}$ explicitly by evaluating the corresponding $\tilde{\Lambda}_{k,\sigma}$ on the $3^D$ elements of the standard basis and (ii) the set $\lambda_D$ is spanning, but is not linearly independent. For example, since $\langle w_1, w_2 \rangle = \langle w_2, w_1 \rangle$, the space of SO(3) invariant functionals on $\mathcal{T}_2 = W^{\otimes 2}$ is one dimensional while $|\lambda_2| = 2$.

*Proof of Proposition 1.* We first show that the bijective correspondence between linear functional $\Lambda : W^{\otimes D} \to \mathbb{R}$ and multi-linear functions $\tilde{\Lambda} : W^D \to \mathbb{R}$, extends to a bijective correspondence between SO(3)-invariant linear/multi-linear functionals. The action of SO(3) on $W^D$ is defined by

$$\tilde{\rho}(R)(w_1, \ldots, w_D) = (Rw_1, \ldots, Rw_D).$$

The action $\rho(R) = R^{\otimes D}$ of SO(3) on $W^{\otimes D}$ is such that the map

$$(w_1, \ldots, w_D) \mapsto w_1 \otimes w_2 \ldots w_D$$

is SO(3)- equivariant. It follows that if $\tilde{\Lambda}$ and $\Lambda$ satisfy equation 24, then for all $R \in$ SO(3), the same equation holds for the pair $\tilde{\Lambda} \circ \tilde{\rho}(R)$ and $\Lambda \circ \rho(R)$. Thus SO(3)-invariance of $\tilde{\Lambda}$ is equivalent to SO(3)-invariance of $\Lambda$.

Multi-linear functionals on $W^D$ invariant to $\tilde{\rho}$ are a subset of the set of polynomials on $W^D$ invariant to $\tilde{\rho}$. It is known (see (Kraft & Procesi, 2000), page 114), that all such polynomials are algebraically generated by functions of the form

$$\det(w_{i_1}, w_{i_2}, w_{i_3}) \text{ and } \langle w_{j_1}, w_{j_2} \rangle, \text{ where } i_1, i_2, i_3, j_1, j_2 \in [D].$$

Equivalently, SO(3)-invariant polynomials are spanned by linear combinations of polynomials of the form

$$\det(w_{i_1}, w_{i_2}, w_{i_3}) \det(w_{i_4}, w_{i_5}, w_{i_6}) \dots \langle w_{j_1}, w_{j_2} \rangle \langle w_{j_3}, w_{j_4} \rangle \dots. \tag{27}$$

When considering the subset of multi-linear invariant polynomials, we see that they must be spanned by polynomials as in equation 27, where each $w_1, \dots, w_D$ appears exactly once in each polynomial in the spanning set. These precisely correspond to the functions in $\lambda_D$.

$\square$

## H    EXPERIMENTS

Table 1: Results obtained on the QM9 dataset Ramakrishnan et al. (2014) for different design choices in the TFN architecture. Results are reported for the $\epsilon_{homo}$ target variable, and are multiplied by $10^3$.

| Model variant | Mean $\ell_1$ error |
|---|---|
| Order 0 irreps | $111.2 \pm 0.3$ |
| Order 0-1 irreps | $82.1 \pm 0.4$ |
| Order 0-2 irreps | $62.1 \pm 1.5$ |
| Order 0-3 irreps | $53.2 \pm 1.9$ |
| Order 0-4 irreps | $51.2 \pm 0.4$ |
| Order 0-3 irreps without non-linearity | $53.1 \pm 0.8$ |
| Order 0-3 irreps + self-interaction only in the final layer | $55.2 \pm 0.3$ |

This section provides an experimental evaluation of different design choices of the TFN architecture, inspired by our theoretical analysis. We study the following questions:

1. **The importance of non-linear activation.** Our proof shows that using non-linear activation functions is not necessary for proving universality. Here, we empirically test the possibility of removing these layers.

2. **The importance of high-dimensional irreducible representations.** Our theoretical analysis shows that in order to represent/approximate high degree polynomials, high-order representations should be used. Here, we check whether using high-order representations has practical benefits.

3. **The effect of self-interaction layers.** Our proof suggests that it is enough to use self interaction linear layers at the end of the model. We empirically compare this approach with the more common approach of using self-interaction layers after each convolutional layer.

**Dataset.**    We use the QM9 (Ramakrishnan et al., 2014) dataset for our experiments. The dataset contains 134K molecules, with node 3D positions, 5 categorical node features and 4 categorical edge features. The task is a molecule property prediction regression task.

**Framework.**    We used pytorch (Paszke et al., 2017)as the deep learning framework and the Deep Graph Library (DGL) (Wang et al., 2019a) as the graph learning framework. All experiments ran on NVIDIA GV100 GPUs.

**Experimental setup.**    We use the the TFN implementation from Fuchs et al. (2020). We trained each model variant for 50 epochs on the $\epsilon_{homo}$ target variable using an $\ell_1$ loss function and the ADAM optimizer with learning rate $10^{-3}$ and report results on the test set on the final epoch averaged over two runs. We used the default parameters and data splits from Fuchs et al. (2020).

**Architecture.** The architecture consists of 4 TFN convolutional layers, each followed by a linear self-interaction layer. We used 16 copies of each irreducible representation used. We used norm-based non-linearity as in the original TFN paper (Thomas et al., 2018). These convolutional layers are followed by a max-pooling layer and two fully connected layers with $16d$ features in the hidden layer, where $d$ is the maximal degree of irreducible representations used.

**Results.** Table 1 and Figure 1 present the results. The main conclusions are: (1) The experiments show that, at least for this task, using non-linear activations does not improve performance. This result fits our theoretical analysis which shows that these layers are not needed for universality. (2) Figure 1 presents a plot of error vs representation degrees used. The plot clearly shows that using high-dimensional representations (up to order 3) improves performance, which also fits our analysis. Using representation orders higher than 3 is significantly more time consuming, and was found to have little effect on the results (as in in Fuchs et al. (2020)), though we believe this to be application-dependent. (3) Using self interaction layers only at the end of the model is shown to have marginal negative effect on the results.

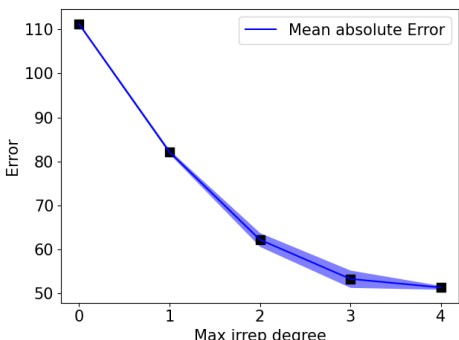

Figure 1: $\ell_1$ error versus maximal irreducible representation used. It is clear that the error is reduced as higher order representation are used. Semi-transparent color represents standard deviation.

