# OpenReview forum: "On the Universality of Rotation Equivariant Point Cloud Networks"
_ICLR.cc/2021/Conference — ICLR 2021 Poster_

### Official Review · AnonReviewer4 · 2020-10-28
**A pedagogical piece with practical relevance**

**Rating:** 8
**Confidence:** 3

**Review:**

**Summarize what the paper claims to contribute.**
The authors claim to: (1) introduce a general approach for proving universality of rotation-translation-permutation equivariant models for point clouds; (2) prove universality of two recent rotation equivariant point cloud networks and (3) introduce two rotation equivariant architectures for point cloud processing

**Strengths:**
The paper is well organized and both the language and notation are clear
The authors consider equivariant representation learning which is of growing interest to the community
The authors present theoretical insights for the success of recent architectures
The insights themselves are leveraged to support the introduction of two new architectures

**Weaknesses:**
The novel architectures are described but not evaluated; it is therefore unclear what impact the simplification will have in practice
It might be nice to point out rotation equivariant architecture that is not universal

**Clearly state your recommendation.**
Accept. See Strengths.

**Arguments for your recommendation.**
Theoretical issues in deep learning is a relevant topic area for ICLR. This paper provides a theoretical framework for interpreting the success of deep learning frameworks for equivariant representation learning on point clouds. Moreover, the paper leverages the insights gathered to propose two novel approaches. The paper is written in a clear and accessible way.

**Possible typos:**
(just before Thm. 1) When these two necessary → when these two sufficient
(just after Lemma 3)  linear function → linear functions

**Post rebuttal**
With consideration of the authors' responses to reviewer questions and revisions to the submitted work I have changed my rating to _clear accept_.

---

> ### Author Response · Authors · 2020-11-17
> **answer to reviewer #4**
>
> We thank the reviewer for the time and effort in reviewing our work. We were glad to see the reviewer found our paper relevant and pedagogical. Below we respond to several points that were raised in the review.
>
> Q: “The novel architectures are described but not evaluated; it is, therefore, unclear what impact the simplification will have in practice”
>
> A:  We agree. The simple architectures were only introduced to illustrate theoretical concepts and inspire  future work. See further discussion in our general post.
>
> Q: Typos
>
> A: Fixed. Thanks for pointing them out!

---

### Official Review · AnonReviewer3 · 2020-10-29
**Difficult to understand and lack of evaluation**

**Rating:** 6
**Confidence:** 2

**Review:**

This work is a theoretical paper investigating the sufficient conditions for an equivariant structure to have the universal approximation property. They show that the recent works: Tensor Field networks and Fuchs et al., are universal under the proposed framework. A simple theoretical architecture is presented as another universal architecture.

Pros:
- Achieving rotation equivariance is important to point cloud network as it is the key to improve the expressiveness of point cloud features. Hence the study of the universality of network with such property is important to the community.
- Overall, the proposed proof looks plausible to me.
- A minimal universal architecture is proposed that satisfies the D-spanning property. This provides the theoretical starting point to design a more advanced and complex equivariant point cloud network.

Cons:
- The paper is quite difficult to follow. I'm not an expert in group theory and had a difficult time understanding some of the theorems and proofs. It would be great if the writing can be broken down into more fundamental modules and provide more illustrations.
- In addition, the paper doesn't provide any evaluation of the proposed new universal architectures. Though this is a theoretical paper, it would be nice to show the proposed theory have some practical use. For instance, it would be great to provide a simple implementation of the minimal universal architecture and show it indeed achieves the rotation equivariant features on the point cloud data. That would make this work much stronger and more practical.

Minors:
There are some typos in the inset figure on page 1: "Equivarint" -> "Equivariant".

---

> ### Author Response · Authors · 2020-11-17
> **answer to reviewer #3**
>
> We thank the reviewer for the time and effort in reviewing our work. Below we respond to several points that were raised in the review.
>
> Q: The paper is quite difficult to follow. I'm not an expert in group theory and had a difficult time understanding some of the theorems and proofs. It would be great if the writing can be broken down into more fundamental modules and provide more illustrations.
>
> A: Thank you for this comment. Although the paper is mathematically intensive we made efforts to make it as accessible as possible. In the revision, we added multiple examples and proof ideas to the text. We are happy to hear suggestions for additional illustrations that can help make the paper more readable.
>
> Q: “it would be great to provide a simple implementation of the minimal universal architecture and show it indeed achieves the rotation equivariant features”
>
> A:  The simple architectures were only introduced to illustrate theoretical concepts and inspire future work. The equivariance of the suggested method can be readily proved (similarly to the equivariance proof in the original TFN paper.) and does not have to be verified experimentally. See further discussion in our general post.

---

### Official Review · AnonReviewer2 · 2020-10-31
**This paper mainly explores the representation ability of invariability of a point cloud network from the  theoretical perspective. The universal approximation property for equivariant architectures under shape-preserving transformations is discussed.**

**Rating:** 6
**Confidence:** 2

**Review:**

This paper mainly explores the representation ability of invariability of a point cloud network from the theoretical perspective. The universal approximation property for equivariant architectures under shape-preserving transformations is discussed. First, the authors derived two sufficient conditions for equivariant architectures with the universal approximation property. Then, they examined two methods based on the Tensor Field Network to prove that such a property holds for both of them. At last, the authors propose alternative methods which also satisfy the universal approximation property.

This paper is full of theoretical analysis, which is based on investigating the equivariant polynomials in the group theory. The proofs are similar to the previous works, except that more transformations, including rotation, translation and permutation, are considered together. However, the proof of the rotation equivariance has been discussed in previous works, and the theorems for translation and permutation are much easier than the rotation analysis. Moreover, the proposed method based on TFN is simply modified by the authors, resulting in alternative architectures which also satisfies the universal approximation property. However, this paper fails to provide any numerical experiments to demonstrate the performance and the influence of parameters \theta.

pros:
- provide sufficient conditions for equivariance of shape-preserving architectures to satisfy the universal approximation property
- prove two methods based of Tensor Field Networks that satisfy the universal approximation property
- raise two simple models based on the TFN


cons:
- This paper provides complete poofs to the TFN network theoretically, but lacks auxiliary experimental verification. Therefore, it is difficult to verify the correctness and feasibility of the proofs.
- The authors do not provide experimental results to demonstrate the performance of the proposed new methods.

---

> ### Author Response · Authors · 2020-11-17
> **answer to reviewer #2**
>
> We thank the reviewer for the time and effort in reviewing our work. Below we respond to several points that were raised in the review.
>
> Q: ”The proofs are similar to the previous works, except that more transformations, including rotation, translation, and permutation, are considered together. However, the proof of the rotation equivariance has been discussed in previous works, and the theorems for translation and permutation are much easier than the rotation analysis.”
>
> A: We respectfully disagree. The main challenge in this paper was to prove universality while *simultaneously* dealing with both permutation and rotational symmetries. We had to come up with novel arguments to deal with this combination, and the argument cannot be reduced to just solving separately for rotations, permutations, and translations.
>
> Q: “This paper … lacks auxiliary experimental verification. Therefore, it is difficult to verify the correctness and feasibility of the proofs. The authors do not provide experimental results to demonstrate the performance of the proposed new methods”.
>
> A: As commonly done in the machine learning community the focus of this paper is theoretical and many of the claims presented would be difficult to validate experimentally (at the risk of stating the obvious it is worth noting that mathematical proofs do not require empirical validation).  The simple architectures were only introduced to illustrate theoretical concepts and inspire future work.  See further discussion in our general post.

---

### Official Review · AnonReviewer1 · 2020-11-01
**Clear and useful proof of universality, but more intuition could be given.**

**Rating:** 8
**Confidence:** 3

**Review:**

Summary:
The authors introduce a framework for sufficient conditions for proving universality of a general class of neural networks that operate on point clouds which takes as input a set of coordinates of points and as output a feature for each point, such that the network is invariant to joint translation of the coordinates, equivariant to permutation of the points and equivariant to joint SO(3) transformations of the coordinates and output features of all points. Notably, this class contains Tensor Field Networks (TFN). The authors accomplish this by writing the network as a composition of an equivariant function from a class F_feat and followed by a linear pooling layer. When the F_feat class satisfies a “D-spanning” criterion and the pooling layer is universal, the network is universal. For a simple class of networks and for TFNs, the authors prove D-spanning. Linear universality of the pooling layer follows from simple representation theory.

Strengths:
-	It is useful to know whether prevalent classes of neural networks are universal
-	The authors use a general construction for proving universality of equivariant networks for the point cloud group, rather than being specific to certain architectures.
-	Reading the proofs along with the main text, the argumentation is clear and relatively easy to follow for me as reviewer, unfamiliar with similar universality proofs.

Weaknesses:
-	The paper would benefit from providing more intuition behind the proposed constructions, lemmas and theorems, in particular this holds for: theorem 1 based on the split between the linear universality and D-spanning; the construction of Segol & Lipman (2019) and how this relates to the Q functions and lemma 2
-	In addition to the previous point, the proofs, currently critical for understanding the paper, are given in the appendix, which is not ideal for the self-containedness of the main paper.
-	As the authors of the TFN paper note: in practice not all higher order irreps of the tensor product of the filter and the features are computed. This seems to indicate a big difference between the theoretical analysis - which includes all irreps and thus is computationally intensive even when modelling low order polynomials – and the practical application of TFNs. It would be interesting to know how expressive such practical low order TFNs are. Another difference between the described networks and practical TFN is that in the described networks, all relevant parameters are in the pooling layer, which sums a large number of terms (looking at the proof of lemma 2, exponential in D), while in practical TFNs, the parameters are in the filters.

Recommendation:
The authors proof the useful statement of universality of a prominent class of neural networks, which is why I recommend the acceptance of this paper.

Suggestion for improvement:
-	Make the big picture clearer by providing more intuition.
-	Comment on the differences between the class of networks described and TFNs used in practice.

Minor points/suggestions:
-	P3 Add definition of W^n_T as n direct sums of W_T
-	P3 “where W_feat is a lifted representation of SO(3)”, what does lifted representation here mean? Just any rep?
-	I get a bit confused by the wording in Def 1. Unless I am mistaken, it appears like the quantifiers are reversed. Should it mean “for every polynomial …, there exists f_1, … in F_feat and linear functionals Lambda_1, …, : W_feat -> R ”?
-	Around Eq 5, perhaps the authors could clarify the clarify the domain of the Q functions, which I suppose is Q^r : R^3n -> T_T, where T=||r||_1
-	Around Eq 7, are X_j and x_j the same?
-	In lemma 4, is A_k any linear map or an equivariant linear map?
-	In Appendix B, perhaps a new subsection B.2 would make sense before theorem 1?
-	In the proof of thm 1, it says “p: R^{d \times n} \to W_T”, should that be W_T^n?
-	In the proof of lemma 2, it says “we see that that exists a linear functional”

## Post rebuttal
I thank the authors for their response and revised version, which has been improved notably with the inclusion of the proof ideas. My previous rating still applies.

---

> ### Author Response · Authors · 2020-11-17
> **Answer to reviewer #`1**
>
> We thank the reviewer for the time and effort in reviewing our work. We were happy to see the reviewer found our paper useful.  Below we respond to several points that were raised in the review.
>
> Q: “The paper would benefit from providing more intuition behind the proposed constructions, lemmas and theorems, in particular, this holds for theorem 1 based on the split between the linear universality and D-spanning; the construction of Segol & Lipman (2019) and how this relates to the Q functions and lemma 2”
>
> A: We used the additional page to add intuition and more discussion on the construction lemmas and theorems. In particular we added more intuition for theorem 1 and the split, and added an explanation of the proof idea for lemma 2.
>
> Q: “In addition to the previous point, the proofs, currently critical for understanding the paper, are given in the appendix, which is not ideal for the self-containedness of the main paper.”
>
> A: We would be happy to have the proofs in the main text. As this is infeasible due to space constraints we added proof ideas to all the claims in the paper as a compromise.
>
> Q: “As the authors of the TFN paper note: in practice not all higher-order irreps of the tensor product of the filter and the features are computed. This seems to indicate a big difference between the theoretical analysis - which includes all irreps and thus is computationally intensive even when modeling low order polynomials – and the practical application of TFNs. It would be interesting to know how expressive such practical low order TFNs are.”
>
> A: This is a good question. We added a discussion of this point in the conclusion section. At the same time we would like to note that practical TFN may change in the future and include higher order representations - in analogy to Alexnet and later Resnet who found ways to successfully implement deeper networks than were used previously.
>
> Q: “Another difference between the described networks and practical TFN is that in the described networks, all relevant parameters are in the pooling layer, which sums a large number of terms (looking at the proof of lemma 2, exponential in D), while in practical TFNs, the parameters are in the filters.”
>
> A: We note that  our networks hold parameters in the filters as well. The filters in TFN are composed of two parts: (1) an MLP that operates on the norm of a point (2) A spherical harmonic function. In our proof, we show that using the MLP in (1) to approximate certain polynomials can lead to universality, but it certainly does not mean that we restrict these MLPs to have a specific form. This is mentioned in Appendix F where we list the differences between the models we used and the original TFN paper.
>
> Regarding the exponential dependency on D, this is essentially unavoidable since the dimension of the space invariant polynomials of degree D is exponential in D (or more accurately, in min(n,D) ). The linear dependence on n (the number of points) makes computing all invariant polynomials of a modest degree D feasible.
>
> Q: ”I get a bit confused by the wording in Def 1. Unless I am mistaken, it appears like the quantifiers are reversed. Should it mean “for every polynomial …, there exists f_1, … in F_feat and linear functionals Lambda_1, …, : W_feat -> R ”?”
>
> A: The order of the quantifiers is correct and this condition is stronger than what you suggest: there is a finite number of polynomials f_1,...f_k that can span all D-degree polynomials p (with appropriate p-dependent linear functions).
>
> Q: “Comment on the differences between the class of networks described and TFNs used in practice.”
>
> A: We added a discussion of the differences pointed out by the reviewer in the conclusions section.
>
> Q: Minor points/suggestions.
>
> A: Great suggestions! They were incorporated in the revised version.

---

### Author Response · Authors · 2020-11-17
**general comments to reviewers**

We thank all reviewers for their valuable feedback. We were happy to see that the reviewers found our paper “Clear and useful”, “relatively easy to follow” and a “useful statement of universality of a prominent class of neural networks” (R1), “important to the community” (R3),  “a pedagogical piece with practical relevance” and “clear and accessible” (R4).

The main issues that were raised are: (i) “the paper can benefit from more intuition” and (ii)  “(the paper) lacks auxiliary experimental verification.”  We have made efforts to address the first issue by adding proof ideas and examples throughout the paper in our revision. For the second issue, while this paper focuses on theoretical analysis of methods which have already been implemented independently, we will be happy to add an experimental section focusing on different design choices of the TFN architecture, inspired by our theoretical analysis.  We hope to present at least some of these experiments by the end of the short discussion period, and in any event we will be sure to add these experiments to the final version.

Regarding the implementation of new architectures, we note that our main goal in this paper was understanding the expressive power of existing state of the art architectures. The new architectures were only suggested to (1) illustrate the claims in the paper in a simple setting and (2) offer a starting point for future work aiming to design universal networks (as mentioned by some reviewers). Implementing these new architectures requires a significant engineering effort and we leave it as future work. We have  added a remark to this effect in the conclusions section.

We have uploaded a revised version. The main changes are additional discussions and proof ideas that make the paper more accessible. Changes in the revised version are marked in blue.

---

### Author Response · Authors · 2020-11-23
**Added experimental appendix**


We have uploaded another revision with an experimental appendix, as discussed in the previous post.

The experiments check several design choices motivated by our theoretical analysis. On the task we considered, we find that simplifying TFN networks to have no activation layers, and a single self interaction layer, as in our proofs, does not (significantly) degrade performance, while using high order representations does have a significant positive  effect (at least up to order 3).

---

### Decision · Program_Chairs · 2021-01-07
**Final Decision**

**Decision:**

Accept (Poster)

**Comment:**

The paper presents a theoretical analysis of the expressive power of equivariant models for point clouds with respect to translations, rotations and permutations. The authors provide sufficient conditions for universality, and prove that recently introduced architectures, e.g. Tensor Field Networks(TFN), do fulfil this property.

The submission received positive reviews ; after rebuttal, all reviewers recommend acceptance and highlight the valuable paper modifications made by the authors to clarify the intuitions behind the proofs.

The AC also considers that this paper is a solid contribution for ICLR, which will draw interest for both theoreticians and practitioners in the community. \
Therefore, the AC recommends acceptance.